# HoSNNs: Adversarially-Robust Homeostatic Spiking Neural Networks with Adaptive Firing Thresholds

**Hejia Geng**                                                                                          *hejia@ucsb.edu*
*Department of Electrical and Computer Engineering*
*University of California, Santa Barbara*

**Peng Li** *                                                                                              *lip@ucsb.edu*
*Department of Electrical and Computer Engineering*
*University of California, Santa Barbara*

**Reviewed on OpenReview:** *https://openreview.net/forum?id=UV58hNygne*

## Abstract

While spiking neural networks (SNNs) offer a promising neurally-inspired model of computation, they are vulnerable to adversarial attacks. We present the first study that draws inspiration from neural homeostasis to design a threshold-adapting leaky integrate-and-fire (TA-LIF) neuron model and utilize TA-LIF neurons to construct the adversarially robust homeostatic SNNs (HoSNNs) for improved robustness. The TA-LIF model incorporates a self-stabilizing dynamic thresholding mechanism, offering a local feedback control solution to the minimization of each neuron's membrane potential error caused by adversarial disturbance. Theoretical analysis demonstrates favorable dynamic properties of TA-LIF neurons in terms of the bounded-input bounded-output stability and suppressed time growth of membrane potential error, underscoring their superior robustness compared with the standard LIF neurons. When trained with weak FGSM attacks ($\epsilon = 2/255$), our HoSNNs significantly outperform conventionally trained LIF-based SNNs across multiple datasets. Furthermore, under significantly stronger PGD7 attacks ($\epsilon = 8/255$), HoSNN achieves notable improvements in accuracy, increasing from 30.90% to 74.91% on FashionMNIST, 0.44% to 36.82% on SVHN, 0.54% to 43.33% on CIFAR10, and 0.04% to 16.66% on CIFAR100.

## 1 Introduction

While neural network models have gained widespread adoption across many domains, a glaring limitation of these models has also surfaced — vulnerability to adversarial attacks (Szegedy et al., 2013; Madry et al., 2017). Subtle alterations in the input can trick a well-tuned neural network into producing misleading predictions, particularly for mission-critical applications (Chakraborty et al., 2018). This vulnerability is shared by both artificial neural networks (ANNs) and spiking neural networks (SNNs) (Sharmin et al., 2019; 2020; Ding et al., 2022), and stands in stark contrast to the inherent robustness of biological nervous systems, prompting interesting questions: Why is the human brain immune to such adversarial noise? Can we leverage biological principles to bolster the resilience of artificial networks?

Motivated by these questions, we offer a new perspective that connects adversarial robustness with homeostatic mechanisms prevalent in living organisms. Homeostasis maintains essential regulatory variables within a life-sustaining range (Bernard, 1865; Cooper, 2008; Pennazio, 2009; Jänig, 2022), and is crucial for stabilizing neural activity (Turrigiano & Nelson, 2004), supporting neurodevelopment (Marder & Goaillard, 2006), and minimizing noisy information transfer (Woods & Wilson, 2013; Modell et al., 2015). Although some studies have investigated homeostasis in SNNs, such as the generalized leaky-integrate-and-fire (GLIF)

---

*Corresponding author

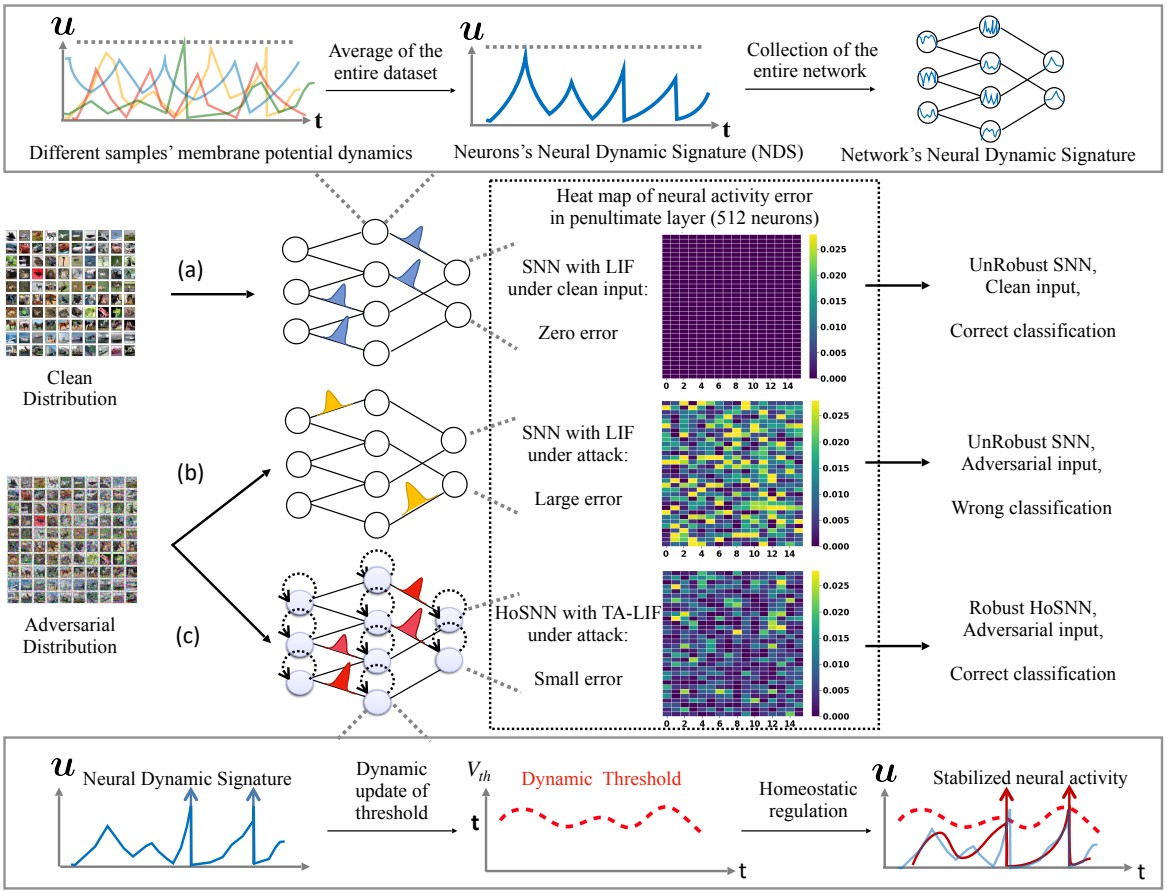

Figure 1: Proposed threshold-adapting leaky integrate-and-fire (TA-LIF) neuron model and homeostatic SNNs (HoSNNs). (a) We leverage a LIF SNN trained using clean data to collect neural dynamic signatures (NDS) as an anchor for the HoSNN (shown in the box above). (b) Adversarial inputs can cause large membrane potential deviations from the NDS in deep layers of LIF SNNs, leading to incorrect model predictions. (c) Homeostatic dynamic threshold voltage control in HoSNNs anchors neural activity based on the NDS, resulting improved robustness.

models (AllenInstitute, 2018; Bellec et al., 2018; 2020; Teeter et al., 2018), no prior work has connected homeostasis with adversarial robustness.

We aim to close this gap by exploring an online biologically-plausible defense solution based on homeostasis. The proposed approach differs from common practices such as adversarial training in a major way, it explicitly builds a localized neural-level self-adapting feedback mechanism into the dynamic operation of the proposed HoSNNs. We view the time-evolving state, i.e., membrane potential $u_i(t|x)$ of each spiking neuron $i$ in a well trained network as its representation of the semantics of the received clean input $x$. Perturbed membrane potential $u_i(t|x')$ resulting from an adversarial input $x'$ corresponds to distorted semantics and can ripple through successive layers to mislead the network output (Li et al., 2021; Rabanser et al., 2019; Nadhamuni, 2021; Shu et al., 2020; Fawzi et al., 2016; Ford et al., 2019; Kang et al., 2019; Ilyas et al., 2019a). For a given pair $(x, x')$, we ensure adversarial robustness by minimizing the total induced membrane potential perturbation $E_i(x, x')$.

We address two practical challenges encountered in formulating and solving this error minimization problem. Firstly, during inference the clean membrane potential $u_i(t|x)$ reference is unknown. As shown in Fig 1(a), we define Neural Dynamic Signature (NDS), the neuron's membrane potential averaged over a given clean

training dataset $\mathcal{D}$, to provide a reference for anchoring membrane potential. Secondly, since externally generated attacks are not known *a priori*, it is desirable to suppress the perturbation of each neuron's membrane potential in an online manner as shown in Fig 1(b). For this, we propose a new threshold-adapting leaky integrate-and-fire (TA-LIF) model with a properly designed firing threshold voltage dynamics that serves as a homeostatic control to suppress undesirable membrane potential perturbations, as shown in Fig 1(c). We theoretically analyze the dynamic properties of TA-LIF neurons in terms of the bounded-input bounded-output stability and suppressed time growth of membrane potential error, underscoring their superior robustness compared with the standard LIF neurons.

We visualize the working of the proposed HoSNNs versus standard LIF-based SNNs in image classification using several CIFAR-10 images in Figure 2. While the adversarial images generated by the Projected Gradient Descent (PGD) (Madry et al., 2017) attack can completely mislead the attention of the LIF SNN, the proposed homeostasis helps the HoSNN focus on parts of the input image strongly correlated with the ground truth class label, leading to significantly improved adversarial robustness as described later.

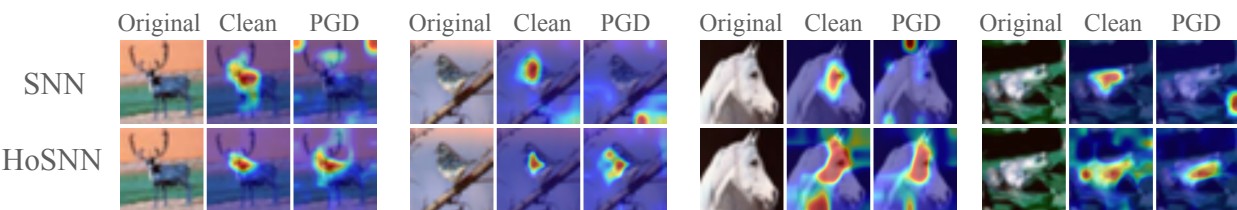

Figure 2: The heatmaps generated by Grad-CAM (Selvaraju et al., 2019) highlight the regions of the input image that most significantly influence the classification decisions of a standard SNN and proposed HoSNN based on the VGG architecture for a set of CIFAR-10 images. The adversarial images are generated using PGD7 with strength of $\epsilon = 6/255$. HoSNN can still maintain attention to the target object under attack.

## 2 Background

### 2.1 Adversarial Attacks

Two notable adversarial attack techniques are the Fast Gradient Sign Method (FGSM) (Goodfellow et al., 2014) and Projected Gradient Descent (PGD) (Madry et al., 2017) method. Let $x$ be an original input, $y$ be the true label, $L(\theta, x, y)$ the loss function with network parameters $\theta$, and $\epsilon$ a small perturbation magnitude, FGSM generates a perturbed input, or an adversarial input example $x'$ by: $x' = x + \epsilon \cdot sign[\nabla_x L(\theta, x, y)]$. PGD is essentially an iterative FGSM. With $x_n$ as the perturbed input in the $n$-th iteration and $\alpha$ as the step size, $Proj_{x+\epsilon}\{\cdot\}$ as the projection within the $\epsilon$-ball of $x$, PGD updates the input by: $x_{n+1} = Proj_{x+\epsilon}\{x_n + \alpha \cdot sign[\nabla_x L(\theta, x_n, y)]\}$. Other gradient-based attacks such as the RFGSM (Tramèr et al., 2017), Basic Iterative Method (BIM) (Kurakin et al., 2018), and DeepFool (Moosavi-Dezfooli et al., 2016) work in a similar fashion, APGD (Croce & Hein, 2020b). Beyond gradient-based methods, a significant area of concern is black-box attacks (Biggio & Roli, 2018).

Some recent studies focused on delivering more powerful attacks in SNNs. Bu et al. (2023) proposed RGA, which leveraged rate-coding in LIF neurons to generate more effective adversarial attacks with a time-extended enhancement. Hao et al. (2024) proposed HART, which leveraged a hybrid gradient calculation that simultaneously incorporates rate-based gradients and timing-based temporal gradients on SNNs.

### 2.2 Defense Methods

Adversarial training is one of the most widely adopted defense methods (Madry et al., 2017), which retrains a model by using a mixture of clean and adversarial examples. Randomization (Xie et al., 2017) introduces stochasticity during inference and can circumvent precise adversarial attacks. The projection technique of (Mustafa et al., 2019) reverts adversarial attacks back to a safer set. Lastly, one may first detect the presence

of adversarial attack and subsequently cope with it (Metzen et al., 2017a). However, these defense methods are not free of limitations (Akhtar & Mian, 2018). Adversarial training relies on precise gradient information and is not biologically plausible (Lillicrap et al., 2020). Randomization, projection, and detection strategies do not fundamentally address the inherent vulnerabilities of ANNs.

## 2.3 Spiking Neuron Models

SNNs allow for spike-based communication and computation (Furber et al., 2014; Gerstner & Kistler, 2002; Deng et al., 2020) and often leverage the Leaky Integrate-and-Fire (LIF) model for each neuron $i$ with the membrane time constant $\tau_m$:

$$\tau_m \frac{du_i(t)}{dt} = -u_i(t) + I_i(t) - \tau_m s_i(t) V_{th}^i(t), \tag{1}$$

where $u_i(t)$ is the membrane potential, $I_i(t) \triangleq R \sum_j w_{ij} a_j(t)$ represents the input and is defined as the sum of the pre-synaptic currents; $w_{ij}$ represents the synaptic weight from neuron $j$ to $i$; $V_{th}^i(t)$ is the firing threshold of neuron $i$ at time $t$. Neuron $i$'s postsynaptic spike train is:

$$s_i(t) = \begin{cases} +\infty & \text{if } u_i(t) \geq V_{th}^i(t) \\ 0 & \text{otherwise} \end{cases} = \sum_f \delta(t - t_i^f) \tag{2}$$

where $\delta(\cdot)$ is the Dirac function, and $t_i^f$ is a postsynaptic spike time. With $\tau_s$ denoting the synaptic time constant, the evolution of the generated postsynaptic current (PSC) $a_j(t)$ is described by:

$$\tau_s \frac{da_j(t)}{dt} = -a_j(t) + s_j(t) \tag{3}$$

The LIF model uses a constant firing threshold. Generalized LIF (GLIF) models employ a tunable threshold with short-term memory, which increases with every emitted output spike, and subsequently decays exponentially back to the baseline threshold (AllenInstitute, 2018; Bellec et al., 2018; 2020; Teeter et al., 2018). However, these models neither consider adversarial robustness nor provide mechanisms to discern "abnormal" from "normal" neural activity.

## 2.4 Spiking Neural Networks Robustness

While there's been growing interest in spiking neural networks (Imam & Cleland, 2020; Pei et al., 2019), empirical studies have demonstrated that SNNs exhibit similar susceptibilities to adversarial attacks (Sharmin et al., 2019; Ding et al., 2022). A line of research has explored porting defensive strategies developed for ANNs to SNNs. To improve the robustness of SNNs, Kundu et al. (2021) proposed a SNN training algorithm jointly optimizing firing thresholds and weights, and Liang et al. (2022) proposed certified training. Ding et al. (2022) enhanced adversarial training by using a Lipschitz constant regularizer. Özdenizci & Legenstein (2023) introduces an adversarially robust ANN-to-SNN conversion algorithm that initializes the SNN with adversarially pre-trained ANN weights, followed by robust fine-tuning. Liu et al. (2024) improved SNN's adversarial robustness by adding a gradient sparsity regularization term in the loss function. However, these methods have not fully addressed the challenges of ensuring adversarial robustness. Additionally, they are computationally expensive and lack biological plausibility.

Another research direction has focused on studying the inherent robustness of SNNs not seen in their ANN counterparts, and factors impacting robustness. Sharmin et al. (2020) recognized the inherent resistance of SNNs to gradient-based adversarial attacks. El-Allami et al. (2021) investigated the impact of key network parameters such as firing voltage thresholds on robustness. Chowdhury et al. (2021) demonstrated the LIF model's noise-filtering capability. Li et al. (2022) explored network inter-layer sparsity. Xu et al. (2022) examined the effects of surrogate gradient techniques on white-box attacks. While these studies have shed light on aspects of SNNs relevant to robustness, effective defense strategies are yet to be developed.

Similar to ours, some studies improve robustness from stability and biological and rational perspectives. Ding et al. (2024a) enhanced adversarial robustness by reducing the mean square of perturbations in the last

neuron layer. Inspired by Stochastic Gating Mechanisms, Ding et al. (2024b) introduced randomness into spike transmission by simulating the probabilistic opening and closing of synaptic and ion channel gates. The fundamental differences between our work and theirs are: 1.We adjust the threshold in a passive and online manner instead of deliberately adding additional optimization loss terms to the loss function as (Ding et al., 2024a), which significantly reduces the additional calculations for training and inference. 2.Our method does not introduce additional randomness as (Ding et al., 2024b). Thus, our method can retain high clean accuracy and ensures that the robustness does not come from gradient obfuscation (Athalye et al., 2018).

## 3 Method

We present the first study that draws inspiration from neural homeostasis to design a threshold-adapting leaky integrate-and-fire (TA-LIF) neuron model and utilize TA-LIF neurons to construct the adversarially robust homeostatic SNNs (HoSNNs) for improved robustness.

### 3.1 Adversarial Robustness as a Membrane Potential Error Minimization Problem

In a well-trained network, we view the time-evolving state, i.e., membrane potential $u_i(t|x)$ of each spiking neuron $i$, over $T$ timesteps as its representation of the semantics of the received clean input $x$. An adversarial input $x' = x + \delta x$, $s.t.$ $\delta x < \epsilon$, where $\epsilon$ is the attack budget (strength), and $\delta x$ is a carefully crafted adversarial noise, may lead to a perturbed membrane potential $u_i(t|x')$, which corresponds to distorted semantics and can ripple through successive layers to mislead the network's decision (Li et al., 2021; Rabanser et al., 2019; Nadhamuni, 2021; Shu et al., 2020; Fawzi et al., 2016; Ford et al., 2019; Kang et al., 2019; Ilyas et al., 2019a).

For a given pair of $(x, x')$, one may ensure adversarial robustness of the network by minimizing the total induced membrane potential perturbation $E_i(x, x')$ of all $N$ neurons over $T$ timesteps:

$$\min E_i(x, x') = \sum_{i=0}^{N} \sum_{t=0}^{T} \parallel u_i(t|x') - u_i(t|x) \parallel^2 \qquad (4)$$

However, there exist two challenges in formulating and solving equation 4. Firstly, since the model is oblivious about the attack, it is impossible to determine how the adversarial input $x'$ is generated, whether it has a corresponding clean input $x$, and what $x$ is if it exists. As such, $u_i(t|x)$ is unknown, which serves as a clean reference in equation 4. We address this problem by inducing the notion of Neural Dynamic Signature (NDS), the neuron's membrane potential averaged over a given clean training dataset $\mathcal{D}$, to provide an anchor for stabilizing membrane potential. Secondly, since externally generated attacks are not known *a priori*, it is desirable to suppress the perturbation of each neuron's membrane potential in an online manner. For this, we propose a new type of spiking neurons, called threshold-adapting leaky integrate-and-fire (TA-LIF) neurons with a properly designed firing threshold voltage dynamics that serves as a homeostatic control to suppress undesirable membrane potential perturbations. We discuss these two techniques next.

### 3.2 Neural Dynamic Signature (NDS) as an Anchor

Eq 4 describes an "ideal" optimization problem. When $E_i(x, x') = 0$, all neuron activities under adversarial sample input are the same as normal sample input, this can certainly achieve "adversarial robustness" in theory, but it is impossible to achieve in practice. There are two main limitations:(1) For a trained network, the intensity and type of attack are determined by external attackers, which means that x' has a huge range of variation. (2) For a trained network, the attack sample x cannot be determined in advance, which means that it is impossible to obtain an accurate $u_i(t|x)$. Under the constraints of these two problems, we still hope to adopt the idea of Eq 4. A feasible approximation is that we use the available average value of the training data set $\mathbb{E}_{x \sim \mathcal{D}} [u_i(t|x)]$ as an approximation of the unavailable $u_i(t|x)$.

We utilize a clean training dataset $\mathcal{D}$ to anchor each spiking neuron $i$. While the membrane potential $u_i(t|x)$ shows variability across individual samples $x$, its expected value over distribution $\mathcal{D}$ denoted by $u_i^*(t|\mathcal{D})$ can act as a reliable reference as illustrated in Fig 1(a):

$$u_i^*(t|\mathcal{D}) \triangleq \mathbb{E}_{x \sim \mathcal{D}}[u_i(t|x)] \qquad (5)$$

The Neural Dynamic Signature (NDS) of neuron $i$ is defined as a temporal series vector over $T$ timesteps: $\boldsymbol{u_i^*}(\mathcal{D}) = [u_i^*(t_1|\mathcal{D}), u_i^*(t_2|\mathcal{D}), \cdots, u_i^*(t_T|\mathcal{D})]$. $\boldsymbol{u_i^*}(\mathcal{D})$ captures the average semantic activation across $\mathcal{D}$. Adversarial perturbations induce input distributional shifts, leading to anomalous activation of out-of-distributional semantics. $\boldsymbol{u_i^*}(\mathcal{D})$ facilitates the identification of neuronal activation aberrations and further offers an anchor signal to bolster network resilience.

We also define the network-level NDS, $\mathcal{U}_{\text{NET}}(\mathcal{D}) \triangleq \{\boldsymbol{u_i}(\mathcal{D})\}_{i=1}^N$, as the collection of NDS vectors in a well-trained LIF-based SNN comprising $N$ neurons. A densely populated SNN with $N$ neurons typically has $O(N^2)$ weight parameters. In contrast, $\mathcal{U}_{\text{NET}}(\mathcal{D})$ scales as $O(NT)$. Recent algorithms have enabled training of high-accuracy SNNs with short latency operating over a small number of time steps, e.g., 5 to 10 (Zhang & Li, 2020). Consequently, the storage overhead of NDS remains manageable.

**Dynamics of NDS**   While serving as an anchor signal, the dynamics of NDS provides a basis for understanding the property of the proposed TA-LIF neurons. As NDS $u_i^*(t|\mathcal{D})$ of neuron $i$ is derived from the well-trained LIF SNN, we take expectation of the LIF dynamic equation 1 with a static firing threshold $V_{th}$ across the entire training distribution $\mathcal{D}$ while simplifying the dynamics by approximating the effects of firing:

$$\tau_m \frac{du_i^*(t|\mathcal{D})}{dt} = -u_i^*(t|\mathcal{D}) + I_i^*(t|\mathcal{D}) - \tau_m r_i^*(\mathcal{D})V_{th} \tag{6}$$

Here, $I_i^*(t|\mathcal{D})$ defines the average current input $\mathbb{E}_{x \sim \mathcal{D}}[I_i(t|\theta, x)]$, and $r_i^*(\mathcal{D})$ denotes the average firing rate $\mathbb{E}_{x \sim \mathcal{D}}[\int_0^{t_T} \frac{s_i(t|x)}{t_T} dt]$.

**Hypothesis of NDS**   We hypothesize that in SNNs, the original neural activity within the training set holds reference significance for mitigating adversarial perturbations. This idea finds support in certain works within the Artificial Neural Network (ANN) domain, which suggest that detecting and modifying neuron activation values in feature space can alleviate adversarial issues (Silva & Najafirad, 2020; Metzen et al., 2017b; Zhang et al., 2020). Some studies deny this view (Carlini & Wagner, 2017; Tramèr, 2022; Ilyas et al., 2019b; Li et al., 2021). In SNNs, whether membrane potential sequences can effectively address adversarial attack problems remains unclear and needs further investigation.

Here we provide an intuitive understanding of NDS's effectiveness. Consider an adversarial sample $x' = x + \delta x$, where the perturbation $\delta x$ represents the adversarial attack. As the strength of the perturbation diminishes ($\delta x \to 0$), the sample gradually converges to the original clean sample ($x' \to x$). Consequently, the neural activity of the attacked network, denoted as $f(x')$, will approach the clean neural activity, $f(x)$, i.e., $f(x') \to f(x)$. This alignment between the attacked and clean neural activity suggests that the network's classification result will likewise revert to the correct prediction, $y' \to y$. Therefore, if we can identify a method to reduce the discrepancy between the neural activity of the attacked network and that of the clean network, we may be able to mitigate the impact of adversarial attacks on the classification outcome.

### 3.3 Threshold-Adapting Leaky Integrate-and-Fire (TA-LIF) Neurons

#### 3.3.1 Membrane Potential Error Minimization with NDS

Instead of examining the deviation of membrane potential $u_i(t|x')$ caused by the adversarial input $x'$ from the unknown $u_i(t|x)$, we define a new error signal $e_i(t|x') \triangleq u_i(t|x') - u_i^*(t|\mathcal{D})$, and replace the optimization problem of equation 4 by a more practical membrane potential error minimization problem while optimizing the dynamically changing firing threshold $V_{th}^i(t|x')$ of each neuron $i$:

$$\min_{\mathbf{V_{th}^i} \in \mathbb{R}^T} E_i(x') = \sum_{t=0}^T e_i(t|x')^2 \triangleq \sum_{t=0}^T (u_i(t|x') - u_i^*(t|\mathcal{D}))^2 \tag{7}$$

where $\mathbf{V_{th}^i} = [V_{th}^i(t_1|x'), V_{th}^i(t_2|x'), \cdots, V_{th}^i(t_T|x')]$. The intrinsic parameter of firing threshold has a critical role in neuronal dynamics and spiking firing. Adapting the firing threshold can mitigate the effects of adversarial noise, and offer an online homeostatic mechanism for minimizing $E_i(x')$, which is potentially generalizable across various attack strengths and types.

### 3.3.2 Error Minimization of TA-LIF Neurons as a Second-Order Homeostatic Control System

Subtracting equation 6 from equation 1 gives the following dynamics of the error $e_i(t)$:

$$\tau_m \frac{de_i(t|x')}{dt} = -e_i(t|x') + \Delta I_i(t|x') - \tau_m[r_i(x')V_{th}^i(t|x') - r_i^*(\mathcal{D})V_{th}] \tag{8}$$

where $\Delta I_i(t|x') \triangleq I_i(t|x') - I_i^*(t|\mathcal{D})$, and $r_i(x') \triangleq \int_0^{t_T} \frac{s_i(t|x')}{t_T} dt$ represents the average firing rate under the adversarial input $x'$. Differentiating equation 8 with respect to time and incorporating the threshold dynamics yields:

$$\tau_m \frac{d^2 e_i(t)}{dt^2} + \frac{de_i(t)}{dt} + r_i\tau_m \frac{dV_{th}^i(t)}{dt} = \varepsilon(t), \tag{9}$$

where $\varepsilon(t) \triangleq \frac{d\Delta I_i(t|x')}{dt}$, and we omit the notational dependencies on $x'$ for clarity. Importantly, equation 9 characterizes the error dynamics $e_i(t)$ as a second-order control system influenced by the external disturbance $\varepsilon(t)$ with $\frac{dV_{th}^i(t)}{dt}$ serving as the control term.

We seek to solve the membrane potential error minimization problem in equation 7 by designing a control scheme that leads to proper error dynamics based on the second-order error system of equation 9. To this end, we utilize control signal $\frac{dV_{th}^i(t)}{dt}$ to provide a negative feedback control to suppress $e_i(t|x')$:

$$\frac{dV_{th}^i(t|x')}{dt} = \theta_i e_i(t|x') = \theta_i[u_i(t|x') - u_i^*(t|\mathcal{D})], \tag{10}$$

where $\theta_i$ is a neuron-level learnable parameter, dictating the pace of firing threshold adjustment.

**TA-LIF model.** equation 1, equation 2, and equation 10 together delineate the proposed TA-LIF model. We construct a homeostatic SNN (HoSNN) using TA-LIF neurons, where each TA-LIF neuron maintains its unique $\theta_i$ and $V_{th}^i(t)$. To extract precise semantic information from $\mathcal{D}$, we collect the NDS for the HoSNN from a well-trained LIF-SNN with identical network configurations. $\theta_i$ and network weights $W$ of the HoSNN can be jointly optimized using a training algorithm such as backpropagation.

During inference with the optimized $\theta_i$, $V_{th}^i(t)$ is adapted in an unsupervised manner according to equation 10. Intuitively, if a TA-LIF neuron $i$ shows a abnormal increased activation relative to the reference NDS, $V_{th}^i(t)$ would be stepped up to suppress the increase in membrane potential. Conversely, if the neuron is abnormally inhibited, $V_{th}^i(t)$ would be tuned down. Figure 3(a) compares the LIF and TA-LIF models via numerical simulation of equation 9, showing the growth of error $e_i(t)$ over time. We set the $\varepsilon(t)$ as Gaussian white noise $\xi(t) \sim \mathcal{N}(0, 1)$ and repeated the simulation 1000 times. The membrane potential error of TA-LIF (red area) is significantly smaller than that of LIF (blue area), revealing TA-LIF's dynamic robustness under noisy input perturbations.

The feedback control in equation 10 offers a straightforward means to implement homeostasis, which in turn enhances the adversarial resilience of the proposed HoSNNs. Furthermore, this homeostatic control exhibits two favorable dynamic properties presented next, underscoring its relevance in solving the membrane potential error minimization problem of equation 7 as a feedback control solution.

### 3.3.3 Theoretical Dynamic Properties of TA-LIF Neurons

We highlight key properties of the TA-LIF dynamics from two perspectives: bounded-input bounded-output (BIBO) stability of membrane potential error and suppressed time growth of error in comparison with the standard LIF model. See **Appendix A** for complete derivation of these properties.

**BIBO Stability** If a system is BIBO stable, then the output will be bounded for every input to the system that is bounded. The characteristic equation and its roots of the proposed second-order TA-LIF dynamics equation 9 incorporating the homeostatic control equation 10 are:

$$\tau_m s^2 + s + r_i\tau_m\theta_i = 0, \; s_{1,2} = \frac{-1 \pm \sqrt{1 - 4r_i\tau_m^2\theta_i}}{2\tau_m} \tag{11}$$

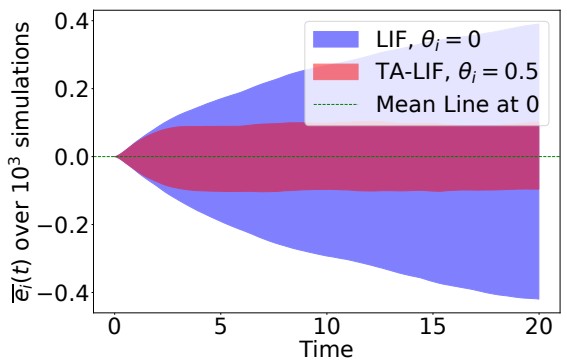

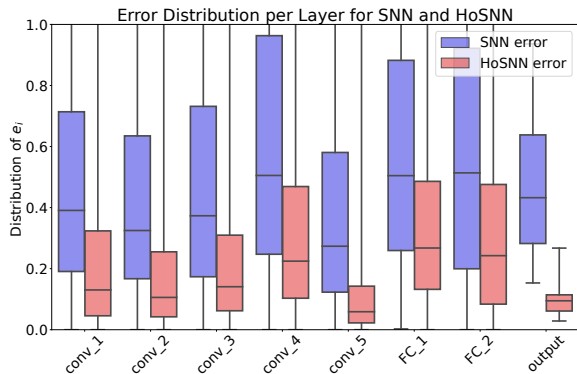

(a) Numerical simulations of equation 9, with parameters $\tau_m = 1$, $r = 1$.

(b) Box plot of the post-synaptic current relative error distribution per layer of a SNN and a HoSNN.

Figure 3: (a) Numerical simulations of equation 9 show that TA-LIF can well suppress the growth of error with time. (b) Box plot of the post-synaptic current relative error distribution per layer of a SNN and a HoSNN, trained with FGSM adversarial training with $\epsilon = 2/255$ and attacked by a same CIFAR10 black-box PGD7 dataset with $\epsilon = 8/255$.

For a second-order system to be BIBO, the roots of its characteristic equation must be a negative real or have a negative real part, which is clearly the case for the TA-LIF model with both $\theta_i$ and $r_i > 0$, affirming the BIBO stability of the TA-LIF model. This signifies that when the adversarial input perturbation $\varepsilon(t)$ is bounded, the deviation of the TA-LIF neuron's membrane potential $e_i(t)$ is also bounded, showing the good control of error under various attack intensities.

**Suppressed Time Growth of Membrane Potential Error** To analyze the evolution of membrane potential error induced by injected input perturbations over time, we follow the common practice (Gerstner et al., 2014; Abbott & Van Vreeswijk, 1993; Brunel, 2000; Renart et al., 2004) to approximate $\Delta I(t)$ in equation 9 as a Wiener process, representing small, independent, and random perturbations. Consequently, the driving force $\varepsilon(t)$ on the right of equation 9 can be approximated by white noise $\xi(t)$ with zero mean and variance $\sigma^2$. By the theory of stochastic differential equations (Kloeden et al., 1992), this leads to the following mean square error for the LIF and TA-LIF models, respectively:

$$\text{LIF}: \frac{dV_{th}^i}{dt} = 0 \implies \langle e_i^2(t) \rangle = O(\sigma^2 t) = \frac{\tau_m^2 \sigma^2}{2} \left( t - \tau_m + \tau_m e^{-t/\tau_m} \right) \tag{12}$$

$$\text{TA-LIF}: \frac{dV_{th}^i}{dt} = \theta_i e_i \implies \langle e_i^2(t) \rangle = O(\sigma^2) = \frac{\tau_m \sigma^2}{2 r_i \theta_i} \left[ 1 - e^{\frac{-t}{2\tau_m}} \left( \cos(\omega_1 t) + \frac{\sin(\omega_1 t)}{2\omega_1 \tau_m} \right) \right] \tag{13}$$

where $\omega_1 = \sqrt{r_i \theta_i - \frac{1}{4\tau_m^2}}$. Importantly, the mean square error of the TA-LIF neuron is $O(\sigma^2)$ and does not grow with time while that of LIF neurons grows unbounded with time. The suppression of time growth of membrane potential error by the TA-LIF model underscores its superiority over the LIF model in terms of adversarial robustness.

### 3.4 Homeostatic SNNs (HoSNNs)

We further introduce the homeostatic SNNs (HoSNNs), which deploy TA-LIF neurons as the basic compute units to leverage their noise immunity. Architecturally, HoSNNs can be constructed by adopting typical connectivity such as dense or convolutional layers, with two learnable parameters: synaptic weights $\boldsymbol{W}$ and threshold dynamics parameter $\boldsymbol{\theta}$ per equation 10. We extract the network-level NDS $\mathcal{U}_{\text{NET}}(\mathcal{D})$ from a LIF-based SNN with identical architecture well-trained on the clean data distribution $\mathcal{D}$. The HoSNNs

optimization problem can be described as:

$$\boldsymbol{W}^*, \boldsymbol{\theta}^* = \arg\min_{\boldsymbol{W}, \boldsymbol{\theta}} \sum_{\{x,y\}} \mathcal{L}_{\text{train}}(x, y \mid \boldsymbol{W}, \boldsymbol{\theta}, \mathbf{V}_{\mathbf{th}}^*(x)) \tag{14}$$

$$\text{s.t.} \mathbf{V}_{\mathbf{th}}^*(x) = \arg\min_{\mathbf{V}_{\mathbf{th}}} \mathcal{L}_{\text{mem}}(\boldsymbol{W}, \boldsymbol{\theta}, \mathbf{V}_{\mathbf{th}}(x), \mathcal{U}_{\text{NET}}(\mathcal{D})|x) \tag{15}$$

where $x$ and $y$ are an input/label pair; $\mathcal{L}_{\text{train}}(\cdot)$ is the loss over a training dataset that can include clean/adversarial examples, or a combination of the two; $\mathcal{L}_{\text{mem}} \triangleq \sum_{i=0}^{N} \sum_{t=0}^{T} e_i(t|x)^2$ is the sum of all neurons' membrane potential error in equation 7. In practice, $\mathcal{L}_{\text{mem}}$ is optimized online by the homeostatic control of firing threshold during the forward process. $\mathcal{L}_{\text{train}}(\cdot)$ is optimized by gradient during the backward process, for which any backpropagation based training algorithm such as BPTT (Neftci et al., 2019; Wu et al., 2018), BPTR (Lee et al., 2020), or TSSL-BP (Zhang & Li, 2020) can be applied to optimize the network based on equation 14.

### 3.5 Complexity Analysis of HoSNN

In a standard Spiking Neural Network (SNN), the number of neurons is $n$, with weights $w = O(n^2)$ and training time $T$. For Higher-order Spiking Neural Networks (HoSNNs), additional space is required to store the learnable parameters $\theta_i = O(n)$, but this storage cost is negligible compared to the network weights. The primary additional cost in training a HoSNN is training the baseline SNN first. The extra time due to the TA-LIF model's dynamic threshold calculation is minimal ($<10\%$). Therefore, the overall training time for a HoSNN is about $2T$. The dynamic threshold adjustment during inference has negligible impact, so the inference cost of a trained HoSNN is almost identical to that of an SNN. There are some challenges in the training of HoSNN. (1) $\theta_i$ Initialization: Initializing $\theta_i$ too large ($> 1$) can destabilize training. It is recommended to set $\theta_i$ between 0 and 0.5, with smaller values needed for complex datasets. (2) Optimizer Choice: HoSNNs converge slower in later training stages, especially in adversarial settings. Using the Adam optimizer with a cosine decay learning rate helps achieve similar convergence speeds as SNNs.

## 4 Experiments

### 4.1 Experimental Setup

The proposed HoSNNs are compared with LIF-based SNNs with identical architecture across four benchmark datasets: Fashion-MNIST (FMNIST) (Xiao et al., 2017), Street View House Numbers (SVHN) (Netzer et al., 2011) CIFAR10 and CIFAR100 (Krizhevsky, 2009). VGG-5/9/11 (Simonyan & Zisserman, 2015) convolutional neural network (CNN) architectures of different sizes and depths are utilized. Widely used methods including FGSM (Goodfellow et al., 2014), RFGSM (Tramèr et al., 2017), PGD (iteration = 7, 20, 40) (Madry et al., 2017), and BIM (Kurakin et al., 2018) are used to generate both white-box and black-box attacks. To test stronger attacks, we introduce APGD (Croce & Hein, 2020a) in our white-box attack, with cross entropy loss (APGD$_{\text{CE}}$), difference of logits ratio loss (APGD$_{\text{DLR}}$) and targeted attack (T-APGD). Unless otherwise specified, PGD refers to PGD7. **The dynamically changing firing thresholds of the HoSNNs are exposed to the attacker and utilized in the gradient calculation when generating white-box attacks.** We independently trained SNNs with the same architecture and used their white-box attacks as the black-box attacks to the HoSNNs.

For each HoSNN, an LIF-based SNN with an identical architecture is trained on the corresponding clean dataset to derive the NDS. Model training employs the BPTT learning algorithm ($T = 5$), leveraging a sigmoid surrogate gradient (Xu et al., 2022; Neftci et al., 2019; Wu et al., 2018). The learning rate for each $\theta_i$ in equation 10, which controls the adaptation of the firing threshold of TA-LIF neurons, is set to 1/10 of that for the network weights, ensuring hyperparameter stability during training. We ensure that $\theta_i$ is non-negative during optimization. We train four types of models: SNNs and HoSNNs on a clean dataset and a weak FGSM-based adversarial training dataset, respectively. For FGSM adversarial training, we set the attack budget to $\epsilon = 2/255$ on FMNIST, SVHN and CIFAR10 as in (Ding et al., 2022) and $\epsilon = 4/255$ on CIFAR100 as in (Kundu et al., 2021). For iterative attacks (PGD & BIM), we adopt

parameters $\alpha = 2.5 * \epsilon / steps$ and $steps = 7, 20, 40$ in accordance with (Ding et al., 2022). Mode experimental settings are in the Appendix C.

## 4.2 Adversarial Robustness under White-box Attacks

**Robustness without adversarial training**  Table 1 compares HoSNNs and SNNs intrinsic resilience without adversarial training. Both types of network are trained exclusively on the clean dataset, and then subjected to $\epsilon = 8/255$ white-box adversarial attacks. The HoSNNs show consistently higher accuracy than the SNN counterparts under all four datasets and four attacks. For example, on CIFAR-10, the HoSNN significantly improves accuracy from 20.86% to 54.76% under FGSM, from 0.54% to 15.32% under PGD7, from 0.69% to 10.35% under APGD with cross entropy loss, from 4.44% to 16.02% under T-APGD.

**Robustness with adversarial training**  In Table 2, we evaluate the enhanced robustness of HoSNNs under adversarial training. We train both the SNNs and HoSNNs using FGSM adversarial training and then expose them to $\epsilon = 8/255$ white-box attacks. The results show a significant boost in robustness for HoSNNs when using adversarial training. Furthermore, the HoSNNs noticeably outperform the SNNs trained using the same FGSM adversarial training. For example, on CIFAR10, the HoSNNs improve the accuracy of the corresponding SNN to 63.98% from 37.93% under FGSM attack, and to 43.33% from 12.42% under PGD7 attack, from 8.23% to 38.89% under APGD-CE attack. On CIFAR100, the HoSNN improves the accuracy to 16.66% from 8.82% under PGD7 attack, from 7.75% to 12.55% under APGD-CE attack.

| Dataset | Net | Clean | FGSM | RFGSM | BIM7 | PGD7 | PGD20 | PGD40 | $APGD_{CE}$ | $APGD_{DLR}$ | T-APGD |
|---|---|---|---|---|---|---|---|---|---|---|---|
| Fashion | × | 92.92 | 56.01 | 70.02 | 38.85 | 30.90 | 28.73 | 27.88 | 23.22 | 40.25 | 39.67 |
| MNIST | ✓ | **92.96** | **65.36** | **76.10** | **49.02** | **35.79** | **32.99** | **32.63** | **37.78** | **61.01** | **50.57** |
| SVHN | × | **95.51** | 26.07 | 42.94 | 2.26 | 0.44 | 0.18 | 0.13 | 0.11 | 7.73 | 0.81 |
|  | ✓ | 93.55 | **44.87** | **57.27** | **12.91** | **4.33** | **1.87** | **1.47** | **3.09** | **12.26** | **5.32** |
| CIFAR | × | **92.47** | 20.86 | 38.72 | 3.29 | 0.54 | 0.38 | 0.3 | 0.69 | 7.03 | 4.44 |
| 10 | ✓ | 92.43 | **54.76** | **62.33** | **28.06** | **15.32** | **11.58** | **10.74** | **10.35** | **27.39** | **16.02** |
| CIFAR | × | **74.00** | 5.74 | 8.94 | 0.10 | 0.04 | 0.01 | 0.00 | 0.00 | 0.01 | 0.01 |
| 100 | ✓ | 71.98 | **13.48** | **12.27** | **0.50** | **0.19** | **0.02** | **0.02** | **2.55** | **0.02** | **0.02** |

Table 1: Training on clean dataset. Whitebox attack results on Fashion-MNIST, SVHN, CIFAR10 and CIFAR100 under various types of attack with an intensity of $\epsilon = 8/255$. The data on the left and right are based on training using the clean and weak FGSM datasets, respectively. HoSNNs (denoted as ✓) provide greater robustness than SNNs (denoted as ×) under all attacks and datasets.

| Dataset | Net | Clean | FGSM | RFGSM | BIM7 | PGD7 | PGD20 | PGD40 | $APGD_{CE}$ | $APGD_{DLR}$ | T-APGD |
|---|---|---|---|---|---|---|---|---|---|---|---|
| Fashion | × | 92.08 | 74.12 | 83.12 | 68.36 | 62.92 | 61.97 | 61.43 | 55.47 | 68.25 | 62.56 |
| MNIST | ✓ | **92.31** | **84.7** | **87.99** | **79.61** | **74.91** | **73.53** | **73.21** | **67.94** | **76.24** | **74.48** |
| SVHN | × | **93.85** | 48.37 | 68.81 | 30.09 | 18.46 | 15.52 | 14.79 | 9.70 | 25.66 | 20.82 |
|  | ✓ | 92.84 | **61.78** | **75.60** | **48.83** | **36.82** | **32.24** | **30.89** | **28.20** | **47.89** | **43.96** |
| CIFAR | × | **91.87** | 37.93 | 59.50 | 22.31 | 12.42 | 11.03 | 10.63 | 8.23 | 16.8 | 14.62 |
| 10 | ✓ | 90.00 | **63.98** | **71.07** | **52.33** | **43.33** | **40.97** | **40.02** | **38.89** | **37.94** | **41.69** |
| CIFAR | × | **68.72** | 22.54 | 36.93 | 13.58 | 8.82 | 7.88 | 7.52 | 7.75 | 7.51 | 5.4 |
| 100 | ✓ | 64.64 | **26.97** | **41.45** | **21.09** | **16.66** | **15.83** | **15.37** | **12.55** | **13.66** | **10.2** |

Table 2: Training on FGSM dataset ($\epsilon = 2/255$). Whitebox attack results on Fashion-MNIST, CIFAR10 and CIFAR100 under various types of attack with an intensity of $\epsilon = 8/255$. The data on the left and right are based on training using the clean and weak FGSM datasets, respectively. HoSNNs (denoted as ✓) provide greater robustness than SNNs (denoted as ×) under all attacks and datasets.

## 4.3 Adversarial Robustness under Black-box Attacks

**Layer-wise Relative Error of Post-synaptic Currents**  Perturbation in membrane potential caused by adversarial inputs can alter the output spike train of each neuron, and the resulting shifts in its post-synaptic current (PSC) can propagate through the successive layers. The PSC $a_i(t)$ is calculated by equation 3. To

reveal the source of HoSNNs robustness, we examine the relative error in PSC induced by adversarial attacks. On CIFAR10, we use the $\epsilon = 8/255$ black-box PGD7 attack to attack SNN and HoSNN trained with the $\epsilon = 2/255$ FGSM adversarial training. For neuron $i$, we record its PSC $a_i(t)$ and $a_i'(t)$ under each pair of clean and adversarial input, respectively, and then calculate the difference of $e_i^{PSC}(t) \triangleq |a_i(t) - a_i'(t)|$ as an error metric. The layer-wise distributions of the relative error $e_i^{PSC}$ are plotted in Figure 3(b) and Figure 4.

The experiment results show that each error distribution of the HoSNN has a significantly reduced mean compared with that of the SNN, and has its probability mass concentrated on low PSC error values, revealing the favorable internal self-stabilization introduced by the proposed homeostasis. Correspondingly, the HoSNN delivers an accuracy of 76.62%, significantly surpassing the 46.97% accuracy of the SNN.

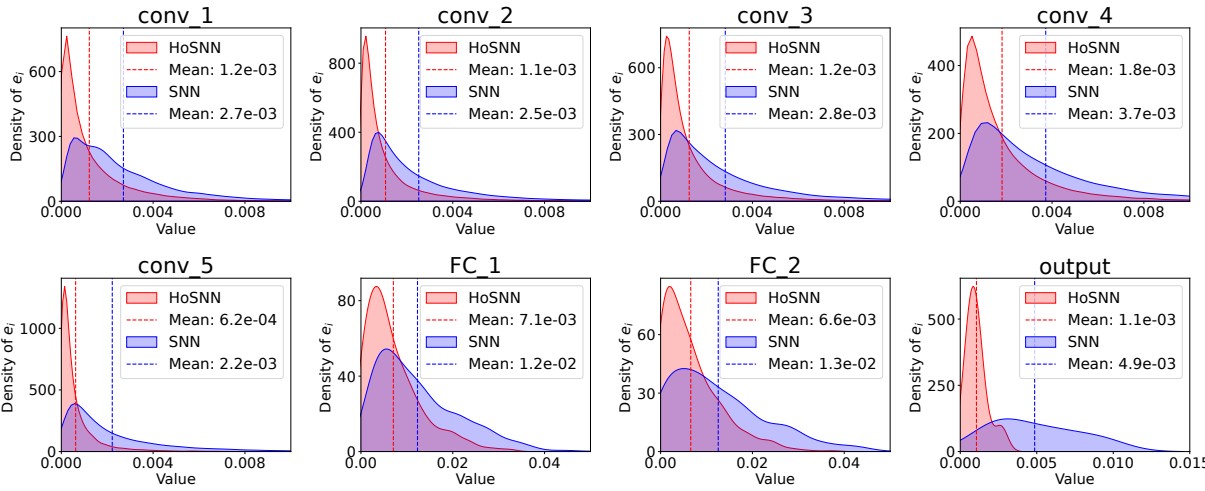

Figure 4: Distribution of post-synaptic current relative error of a SNN and HoSNN trained with FGSM adversarial training with $\epsilon = 2/255$ and attacked by black-box PGD7 attack with $\epsilon = 8/255$.

**Black Box Robustness** Table 3 evaluates the robustness of SNNs (denoted as ×) and HoSNNs (denoted as ✓) against black-box attacks. All models are trained by weak FGSM adversarial training and tested by $\epsilon = 32/255$ black-box attacks generated using separately trained SNNs with identical architecture. Table 3 shows that the HoSNNs exhibit significantly stronger black-box robustness than the SNN counterparts. For example on CIFAR-10, the HoSNN outperforms the traditional SNN under the FGSM and PGD7 attacks with 11.7% and 13.31% accuracy improvements, respectively. On CIFAR-100, HoSNN improves the PGD7 accuracy from 6.74% to 16.90%.

| Dataset | Net | Clean | FGSM | RFGSM | PGD7 | BIM7 |
|---|---|---|---|---|---|---|
| Fashion MNIST | × | 92.08 | 66.26 | 80.09 | 74.08 | 73.43 |
|  | ✓ | **92.31** | **68.31** | **80.73** | **75.12** | **74.09** |
| SVHN | × | **93.85** | 17.37 | 42.64 | 21.16 | 36.70 |
|  | ✓ | 92.84 | **19.08** | **44.57** | **25.97** | **40.75** |
| CIFAR 10 | × | **91.87** | 13.48 | 8.79 | 0.11 | 0.31 |
|  | ✓ | 90.00 | **25.18** | **31.11** | **13.42** | **23.34** |
| CIFAR 100 | × | **68.72** | 12.18 | 17.87 | 6.74 | 17.36 |
|  | ✓ | 64.64 | **14.54** | **24.32** | **16.90** | **32.04** |

Table 3: SNNs and HoSNNs black-box attack accuracy, trained with FGSM adversarial training and tested by different black-box attack methods with $\epsilon = 32/255$.

### 4.4 Checklist for gradient obfuscation

For a detailed check of **gradient obfuscation** (Athalye et al., 2018; Carlini et al., 2019), we provide comprehensive inspection and data (in Appendix D). HoSNNs pass all five tests as show in Table 4

**For Test (1)** We plot the curves of white-box FGSM and PGD7 attacks on four datasets in Figure 5 and Table 8, with attack budgets $\epsilon$ from 0 to $64/255$ to ensure that the network can be completely fooled. The

| Items to identify gradient obfuscation | HoSNN | Experiment |
|---|---|---|
| (1) Single-step attack performs better compared to iterative attacks | ✓ | Fig 5 and Table 8 |
| (2) Black-box attacks perform better compared to white-box attacks | ✓ | Fig 6 and Table 9 |
| (3) Increasing perturbation bound can't increase attack strength | ✓ | Fig 7 and Table 10 |
| (4) Unbounded attacks can't reach 100% success | ✓ | Fig 7 and Table 10 |
| (5) Adversarial example can be found through random sampling | ✓ | Fig 8 |

Table 4: Checklist for gradient obfuscation

red curve is the accuracy under FGSM, while the blue curve is PGD7. From the Figure 5 we can confirm that all iterative attacks are much stronger than single-step attack.

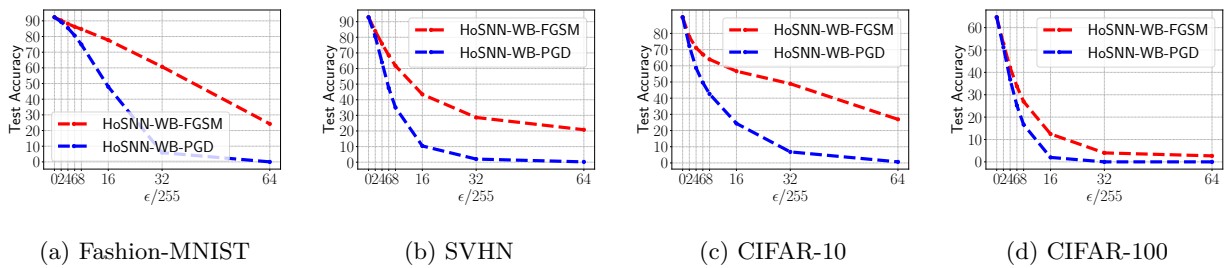

(a) Fashion-MNIST  (b) SVHN  (c) CIFAR-10  (d) CIFAR-100

Figure 5: For Test (1). Performance of HoSNN under white-box FGSM and PGD7 attack.

**For Test (2)** We plot the curves of white-box and black-box PGD7 attacks on four datasets in Figure 6 and Table 9, with attack budgets $\epsilon$ from 0 to 64/255 to ensure that the network can be completely fooled. The green curve is the accuracy under black-box PGD7 attack, while the blue curve is under white-box. From the Figure 6 we can confirm that all white-box attacks are much stronger than black-box.

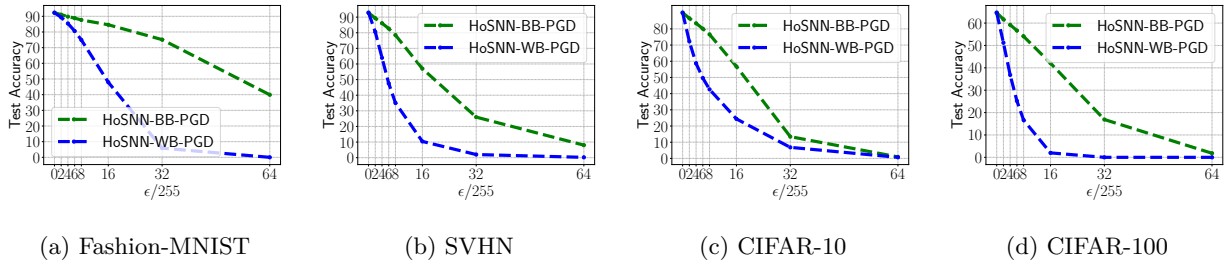

(a) Fashion-MNIST  (b) SVHN  (c) CIFAR-10  (d) CIFAR-100

Figure 6: For Test (2). Performance of HoSNN under white-box and black-box and PGD7 attack.

**For Test (3) & (4)** We increasing perturbation bound can't increase attack strength & Unbounded attacks can't reach ∼ 100% success, we plot the curves of white-box and black-box PGD7 attacks on four datasets in Figure 7 and Table 10, with attack budgets $\epsilon$ from 0 to 64/255 to ensure that the network can be completely fooled. The blue curve is HoSNN's accuracy under white-box PGD7 attack, while the yellow curve is SNN's baseline. From the Figure 7 we can confirm that as perturbation bound increasing HoSNN's accuracy is decreasing and all unbounded attacks reach ∼ 100% success.

**For Test (5)** Since all gradient-based attacks work, there is no need to use random sampling methods. Therefore Test (5) passed obviously. Figure 8 shows model performance with and without adversarial training (ADV) under white-box PGD7 attacks with varying intensities: $\epsilon = 2, 4, 6, 8/255$. The proposed HoSNNs consistently outperform the SNNs in terms of model accuracy across all attack intensities and four datasets.

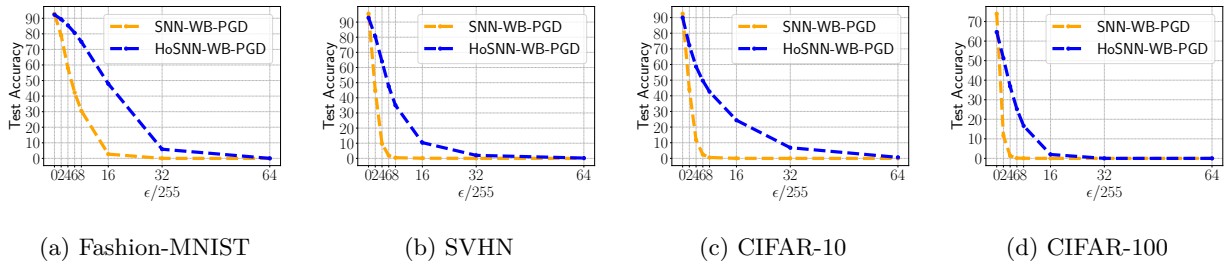

(a) Fashion-MNIST  (b) SVHN  (c) CIFAR-10  (d) CIFAR-100

Figure 7: For Test(3) and (4). Performance of HoSNN under larger white-box PGD7 attack.

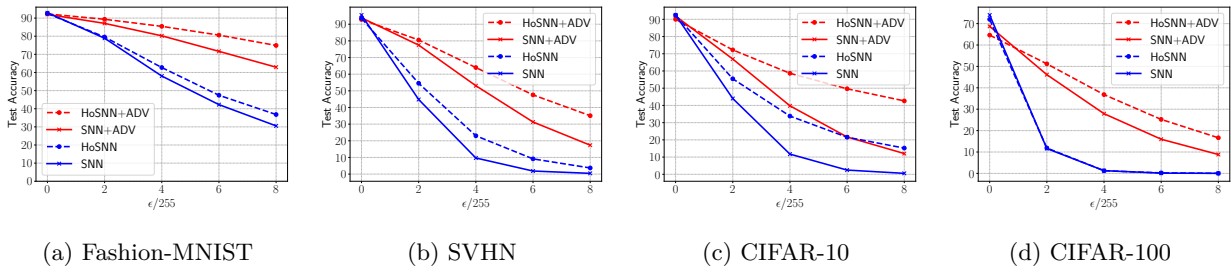

(a) Fashion-MNIST  (b) SVHN  (c) CIFAR-10  (d) CIFAR-100

Figure 8: For Test(5). Performance of SNNs and HoSNNs as a function of white-box PGD7 attack intensity.

## 4.5 Comparison with other works

We compare our method against recent state-of-the-art defense methods under white-box FGSM and PGD7 attacks in Table 5. We tested all three methods on the same VGG7 network architecture, detailed in Appendix A. We use $\epsilon = 2$ FGSM adversarial training for CIFAR10 and $\epsilon = 4$ for CIFAR100. Adversarial examples are generated from the gradient of BPTT. Our adversarially trained models achieved the highest defense accuracies.

| Data | Methods | Clean | FGSM | PGD7 |
|------|---------|-------|------|------|
| CIFAR-10 | Ding et al. (2022) | 83.45 | 39.69 | 20.14 |
| | Özdenizci & Legenstein (2023) | **91.86** | 41.55 | 27.35 |
| | Our work | 90.00 | **63.98** | **42.63** |
| CIFAR-100 | Ding et al. (2022) | 67.47 | 25.38 | 15.66 |
| | Özdenizci & Legenstein (2023) | **67.26** | 21.35 | 13.45 |
| | Our work | 64.64 | **26.97** | **16.66** |

Table 5: Comparison with others work. We use $\epsilon = 2$ FGSM adversarial training for CIFAR10 and $\epsilon = 4$ for CIFAR100. Adversarial examples are generated from the gradient of BPTT.

## 5 Discussions

This paper presents the first work on biologically inspired homeostasis for enhancing adversarial robustness of spiking neural networks. Specifically, we propose a new TA-LIF model with a threshold adaptation mechanism and use TA-LIF neurons to construct inherently more robust HoSNN networks. Yet, there is room for future investigations including better trading off between model accuracy under clean and adversarial inputs. More broadly, we recognize the vast and yet untapped potential of biological homeostasis in neural network research. The relationship between the properties of individual neurons and the overall performance of the network warrants further exploration.

## 6 Acknowledgement

This material is based upon work supported by the National Science Foundation under Grants No. 1948201 and No. 2310170.

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

# A  Appendix

We mainly present the derivation of the second-order dynamic equation of TA-LIF in B, dynamic stability analysis in C the detailed experimental setup in D and checklist for gradient obfuscation in E.

# B  Derivation of TA-LIF Dynamic Equation

In this section, we derive the approximate second-order dynamic equations of the threshold-adapting leaky integrate-and-fire (TA-LIF) neurons and subsequently analyze them.

## B.1  LIF Dynamics

To facilitate our discussion, let's commence by presenting the first-order dynamic equations of the LIF neuron $i$ at time $t$:

$$\tau_m \frac{du_i(t)}{dt} = -u_i(t) + I_i(t) - \tau_m s_i(t) V_{th} \tag{16}$$

The input current is defined as

$$I_i(t) = R \sum_j w_{ij} a_j(t) \tag{17}$$

The spiking behavior $s_i(t)$ is defined as:

$$s_i(t) = \begin{cases} +\infty & \text{if } u_i(t) \geq V_{th} \\ 0 & \text{otherwise} \end{cases} = \sum_f \delta(t - t_i^f) \tag{18}$$

And the post-synaptic current dynamics are given by:

$$\tau_s \frac{da_j(t)}{dt} = -a_j(t) + s_j(t) \tag{19}$$

Where:

- $\tau_m$: Represents the membrane time constant.

- $I_i(t)$: Denotes the input, which is the summation of the pre-synaptic currents.

- $w_{ij}$: Stands for the synaptic weight from neuron $j$ to neuron $i$.

- $a_j(t)$: Refers to the post-synaptic current induced by neuron $j$ at time $t$.

- $V_{th}$: Is the static firing threshold.

- $t_i^f$: Indicates the $f$-th spike time of neuron $i$.

- $\tau_s$: Is the synaptic time constant.

## B.2 Neural Dynamic Signature

Let's begin by reviewing the definition of the Neural Dynamic Signature (NDS). Given a data instance $x$ sampled from distribution $\mathcal{D}$, the NDS of neuron $i$, contingent upon the training set distribution $\mathcal{D}$, can be represented as a temporal series vector $\boldsymbol{u_i^*}([\mathcal{D})$. Specifically, at time $t$, it holds the value:

$$u_i^*(t|\mathcal{D}) \triangleq \mathbb{E}_{x\sim\mathcal{D}}[u_i(t|x)], \text{ for } t \in [0, T] \tag{20}$$

To derive the dynamics of NDS, we start by revisiting Equation equation 16, rewriting it with respect to $x$

$$\tau_m \frac{du_i(t|x)}{dt} = -u_i(t|x) + I_i(t|x) - \tau_m s_i(t|x)V_{th} \tag{21}$$

For the convenience of dynamic analysis, we choose to approximate the discontinuous Dirac function term $s_i(t|x)$ with the average firing rate $r_i(x)$ of neuron $i$. The average firing rate is calculated by

$$r_i(x) \triangleq \int_0^T \frac{s_i(t|x)}{T} dt \tag{22}$$

We take this approximation to address discontinuity of Dirac funtion, as $r_i(x)$ and $s_i(t|x)$ have the same integral value over time: the number of neuron firings. Substituting Equation equation 22 into Equation equation 21 and computing the expectation on both sides, we have:

$$\tau_m \mathbb{E}_{x\sim\mathcal{D}}[\frac{du_i(t|x)}{dt}] = -\mathbb{E}_{x\sim\mathcal{D}}[u_i(t|x)] + \mathbb{E}_{x\sim\mathcal{D}}[I_i(t|x)] - \tau_m \mathbb{E}_{x\sim\mathcal{D}}[r_i(x)]V_{th} \tag{23}$$

Here, we denote the average input current of neuron $i$ over the entire dataset as:

$$I_i^*(t|\mathcal{D}) \triangleq \mathbb{E}_{x\sim\mathcal{D}}[I_i(t|x)] \tag{24}$$

and the average spike frequency of neuron $i$ over the entire dataset as:

$$r_i^*(\mathcal{D}) \triangleq \mathbb{E}_{x\sim\mathcal{D}}[r_i(x)] \tag{25}$$

With the definition from Equation equation 20, the dynamics of NDS can be expressed as:

$$\tau_m \frac{du_i^*(t|\mathcal{D})}{dt} = -u_i^*(t|\mathcal{D}) + I_i^*(t|\mathcal{D}) - \tau_m r_i^*(\mathcal{D})V_{th} \tag{26}$$

As mentioned in the main text, we usually expect NDS to have precise semantic information of the distribution $\mathcal{D}$. So NDS should be obtained through a well-trained model with optimal weight parameter $\boldsymbol{W^*}$. For clarity in the following sections, we use $\boldsymbol{W^*}$ to represent the actually used NDS:

$$\tau_m \frac{du_i^*(t|\boldsymbol{W^*}, \mathcal{D})}{dt} = -u_i^*(t|\boldsymbol{W^*}, \mathcal{D}) + I_i^*(t|\boldsymbol{W^*}, \mathcal{D}) - \tau_m r_i^*(\boldsymbol{W^*}, \mathcal{D})V_{th} \tag{27}$$

### B.3 TA-LIF Dynamics

In this section, we delve deeper into the dynamical equations governing the TA-LIF neuron and derive its second-order dynamic equation

$$\tau_m \frac{du_i(t)}{dt} = -u_i(t) + I_i(t) - \tau_m s_i(t) V_{th}^i(t) \tag{28}$$

The synaptic input $I_i(t)$, the spike generation function $s_i(t)$ and post-synaptic current dynamics of TA-LIF are defined same as equation 17equation 18equation 19. For a specific network parameter $\boldsymbol{W}$ and a sample $x'$ drawn from $\mathcal{D}'$, the dynamic equation governing the threshold $V_{th}^i(t)$ is:

$$\frac{dV_{th}^i(t|\boldsymbol{W}, x')}{dt} = \theta_i e_i(t|\boldsymbol{W}, x'), \tag{29}$$

where the error signal, utilizing the NDS as given in equation 20, is defined as:

$$e_i(t|\boldsymbol{W}, x') \triangleq u_i(t|\boldsymbol{W}, x') - u_i^*(t|\boldsymbol{W^*}, \mathcal{D}) \tag{30}$$

Applying the continuity approximation for the Dirac function as per equation 22 and incorporating the conditional dependency of $\boldsymbol{W}$ and $x'$, rewriting the dynamics for TA-LIF equation 28 as:

$$\tau_m \frac{du_i(t|\boldsymbol{W}, x')}{dt} = -u_i(t|\boldsymbol{W}, x') + I_i(t|\boldsymbol{W}, x') - \tau_m r_i(\boldsymbol{W}, x') V_{th}^i(t|\boldsymbol{W}, x') \tag{31}$$

Subtracting equation 27 from equation 31 and employing equation 30, denoting

$$\Delta I_i(t|\boldsymbol{W}, x') \triangleq I_i(t|\boldsymbol{W}, x') - I_i^*(t|\boldsymbol{W^*}, \mathcal{D}) \tag{32}$$

we derive the 1st-order dynamic of $e_i(t|\boldsymbol{W}, x')$

$$\tau_m \frac{de_i(t|\boldsymbol{W}, x')}{dt} = -e_i(t|\boldsymbol{W}, x') + \Delta I_i(t|\boldsymbol{W}, x') - \tau_m [r_i(\boldsymbol{W}, x') V_{th}^i(t|\boldsymbol{W}, x') - r_i^*(\boldsymbol{W^*}, \mathcal{D}) V_{th}] \tag{33}$$

Differentiating equation 33 with respect to time and utilizing the threshold dynamics from equation 30, we obtain:

$$\tau_m \frac{d^2 e_i(t|\boldsymbol{W}, x')}{dt^2} = -\frac{de_i(t|\boldsymbol{W}, x')}{dt} + \frac{d\Delta I_i(t|\boldsymbol{W}, x')}{dt} - \tau_m \theta_i r_i(\boldsymbol{W}, x') e_i(t|\boldsymbol{W}, x') \tag{34}$$

For succinctness, we will omit dependencies on $\boldsymbol{W}$ and $x'$, resulting in TA-LIF dynamics in the main text equation 26:

$$\tau_m \frac{d^2 e_i(t)}{dt^2} + \frac{de_i(t)}{dt} + r_i \tau_m \theta_i e_i(t) = \frac{d\Delta I_i(t)}{dt} \tag{35}$$

For the standard LIF neurons where $\theta_i \to 0$, the equation simplifies to:

$$\tau_m \frac{d^2 e_i(t)}{dt^2} + \frac{de_i(t)}{dt} = \frac{d\Delta I_i(t)}{dt} \tag{36}$$

## C   Dynamic Stability Analysis

In this section, we analyze the stability of equation 35 and equation 36 to explore the influence of our dynamic threshold mechanism on the noise suppression ability of the TA-LIF neuron.

## C.1 BIBO Stability of Equation equation 35

*Characteristic Equation:* We first show the BIBO (Bounded Input, Bounded Output) stability Ogata (2010) of TA-LIF neurons based on equation 35. The characteristic equation of equation 35 of non-silent ($r_i > 0$) and non-degenerating ($\tau_m, \theta_i > 0$) TA-LIF neurons is:

$$\tau_m s^2 + s + r_i \tau_m \theta_i = 0 \tag{37}$$

and its roots are

$$s_{1,2} = \frac{-1 \pm \sqrt{\Delta}}{2\tau_m}, \ \Delta = 1 - 4r_i \tau_m^2 \theta_i \tag{38}$$

- For $\Delta > 0$: Both roots $s_{1,2}$ are real and negative.

- For $\Delta = 0$: There's a single negative real root.

- For $\Delta < 0$: Both roots are complex with negative real parts.

For a second-order system to be BIBO, the roots of its characteristic equation must be negative real or have negative parts, which is clearly the case for the TA-LIF model under the above three situations, affirming the BIBO stability of equation 35. The BIBO stability signifies that with the bounded driving input to system equation 35, the deviation of the TA-LIF neuron's membrane potential from its targeted NDS is also bounded, demonstrating the well control of the growth of error $e_i(t)$.

## C.2 Stability of Equation equation 35 Under White Noise

To elucidate the dynamic characteristics of TA-LIF further, we adopt the prevalent method Abbott & Van Vreeswijk (1993); Brunel (2000); Gerstner et al. (2014); Renart et al. (2004), approximating $\Delta I(t)$ with a Wiener process. This approximation effectively represents small, independent, and random perturbations. Hence, the driving force in equation equation 35 $\frac{d\Delta I_i(t)}{dt}$ can be modeled by a Gaussian white noise $F(t)$, leading to the well-established Langevin equation in stochastic differential equations theory Kloeden et al. (1992); Van Kampen (1992); Risken (1996):

$$\frac{d^2 e_i(t)}{dt^2} + \frac{1}{\tau_m} \frac{de_i(t)}{dt} + r_i \theta_i e_i(t) = F(t) \tag{39}$$

Denoting $\langle \cdot \rangle$ as averaging over time, $F(t)$ is a Gaussian white noise with variance $\sigma^2$ that satisfies:

$$\begin{cases} \langle F(t) \rangle & = 0, \\ \langle F(t_1) F(t_2) \rangle & = \sigma^2 \delta(t_1 - t_2), \\ \langle F(t_1) F(t_2) \cdots F(t_{2n+1}) \rangle & = 0, \\ \langle F(t_1) F(t_2) \cdots F(t_{2n}) \rangle & = \sum_{\text{all pairs}} \langle F(t_i) F(t_j) \rangle \cdot \langle F(t_k) F(t_l) \rangle \cdots \end{cases} \tag{40}$$

where the sum has to be taken over all the different ways in which one can divide the $2n$ time points $t_1 \cdots t_{2n}$ into $n$ pairs. Under this assumption equation 40, the solution of the Langevin equation equation 39 is Uhlenbeck & Ornstein (1930); Wang & Uhlenbeck (1945):

$$\langle [\Delta e_i(t)]^2 \rangle = \frac{\tau_m \sigma^2}{2r_i \theta_i} \left[ 1 - e^{\frac{-t}{2\tau_m}} \left( \cos(\omega_1 t) + \frac{\sin(\omega_1 t)}{2\omega_1 \tau_m} \right) \right] = O(\sigma^2) \tag{41}$$

where $\Delta e_i(t) = e_i(t) - \langle e_i(t) \rangle$ and $\omega_1 = \sqrt{r_i \theta_i - \frac{1}{4\tau_m^2}}$. While under the same assumptions equation 40, equation equation 36 yields:

$$\langle [\Delta e_i(t)]^2 \rangle = \frac{\tau_m^2 \sigma^2}{2} \left( t - \tau_m + \tau_m e^{-t/\tau_m} \right) = O(\sigma^2 t) \tag{42}$$

Obviously, Gaussian white noise with zero mean equation 40 leads $\langle e_i(t) \rangle = 0$, $\langle [\Delta e_i(t)]^2 \rangle = \langle e_i^2(t) \rangle$. Hence,

$$\frac{d\Delta I(t)}{dt} \sim F(t) \implies \begin{cases} \langle e_i^2(t) \rangle_{LIF} = O(\sigma^2 t) \\ \langle e_i^2(t) \rangle_{TA-LIF} = O(\sigma^2) \end{cases} \tag{43}$$

Significantly, the mean square error $\langle e_i^2(t) \rangle_{TA-LIF}$ of the TA-LIF neuron remains bounded to $O(\sigma^2)$ and doesn't increase over time. In contrast, under identical input perturbations, the mean square error $\langle e_i^2(t) \rangle_{LIF}$ of the LIF neuron may grow unbounded with time, highlighting its potential susceptibility to adversarial attacks.

### C.3  A more intuitive explanation

We would like to provide a more intuitive explanation of Eq 9 and Eq 10. Notice that when $\theta_i = 0$ according to Eq 10, TA-LIF no longer has a dynamically changing threshold and thus degenerates into standard LIF. Its second-order dynamics degenerates into:

$$\tau_m \frac{d^2 e_i(t)}{dt^2} + \frac{d e_i(t)}{dt} = \varepsilon(t)$$

An example to intuitively understand these two formulas is that they correspond to a driven damped oscillator in the physical world, that is, the movement of a ball connected to a spring under a driving force and damping environment. Considering $e_i(t)$ as the distance the ball deviates from the equilibrium position, we can understand the physical meaning of each term in the formula: $\tau_m \frac{d^2 e_i(t)}{dt^2}$ describes the acceleration of the ball; $\frac{d e_i(t)}{dt}$ describes the frictional resistance of the ball; $r_i \tau_m \theta_i e_i(t)$ describes the restoring force of the spring, which is the tendency of the spring to pull the ball toward the equilibrium position ($e_i(t) = 0$); $\varepsilon(t)$ describes the driving force exerted by the environment on the ball.

With this explanation, we can easily understand the difference between the second-order dynamics of TA-LIF and LIF. The key term is $r_i \tau_m \theta_i e_i(t)$. For TA-LIF, it is like a small ball that is constantly pulled toward its initial equilibrium position by a spring when it is subjected to external disturbances: the result is that the ball will not deviate too far ($e_i(t)$ remains small). However, there is no such term in the dynamics of LIF, which causes $e_i(t)$ to become larger under external disturbances.

## D  Experiment Setting Details

Our evaluation encompasses three benchmark datasets: FashionMNIST, SVHN, CIFAR10, and CIFAR100. For experimental setups, we deploy:

- LeNet (32C5-P-64C5-P-1024-10) for FashionMNIST.

- VGGs (32C3-32C3-P-64C3-P-128C3-128C3-128-10) for SVHN.

- VGGs (128C3-P-256C3-P-512C3-1024C3-512C3-1024-512-10) for CIFAR10.

- VGGs (128C3-P-256C3-P-512C3-1024C3-512C3-1024-1024-100) for CIFAR100.

Here, the notation 32C3 represents a convolutional layer with 32 filters of size $3 \times 3$, and P stands for a pooling layer using $2 \times 2$ filters. For the CIFAR10 and CIFAR100 datasets, we incorporated batch normalization layers and dropout mechanisms to mitigate overfitting and elevate the performance of the deep networks. In our experiments with FashionMNIST, SVHN, and CIFAR10, the output spike train of LIF neurons was retained to compute the kernel loss, as described in Zhang & Li (2020). For CIFAR100, we directly employed softmax for performance.

For all HoSNN experiments, a preliminary training phase was carried out using an LIF SNN, sharing the same architecture, on the clean datasets to deduce the NDS. Hyperparameters for LIF and TA-LIF neurons

included a simulation time $T = 5$ , a Membrane Voltage Constant $\tau_m = 5$, and a Synapse Constant $\tau_s = 3$. For the TA-LIF results in the main text, we assigned $\theta_i$ initialization values of 5 for FashionMNIST, SVHN, CIFAR10 and 3 for CIFAR100. All neurons began with an initial threshold of 1. The step function was approximated using $\sigma(x) = \frac{1}{1+e^{-5x}}$, where $x = u(t) - V_{th}(t)$ and the BPTT learning algorithm was employed. For TA-ALIF neurons, the learning rate for $\theta_i$ was set at 1/10 of the rate designated for weights, ensuring hyperparameter stability during training. We also constrained $\theta_i$ to remain non-negative during optimization, ensuring a possible transition from TA-LIF to LIF. For the generation of all gradient-based adversarial attacks, we assume that the attacker can know all $V_{th}(t)$ and use it in the gradient, even if it is generated dynamically and is not stored as network parameters. During training, we use $V_{th}(t) = V_{th}(0)$. We utilized the Adam optimizer with hyperparameters betas set to (0.9, 0.999), and the $lr = 5 \times 10^{-4}$ with cosine annealing learning rate scheduler ($T = $ epochs). We set batch size to 64 and trained for 200 epochs. All images were transformed into currents to serve as network input. Our code is adapted from Zhang & Li (2020). The experiment used four NVIDIA A100 GPUs. For CIFAR10 and CIFAR100, it took up to about 48 hours for adversarial training.

Regarding adversarial attack, we use an array of attack strategies, including FGSM, RFGSM, PGD, and BIM. For both CIFAR10 and CIFAR100, we allocated an attack budget with $\epsilon = 8/255$. For iterative schemes like PGD, we set $\alpha = 2.5 * \epsilon/steps$ and $steps = 7, 20, 40$, aligning with the recommendations in Ding et al. (2022). For the adversarial training phase, FGSM training was used with $\epsilon$ values of 2/255 for CIFAR10 as per Ding et al. (2022) and 4/255 for CIFAR100, following Kundu et al. (2021).

| Items to identify gradient obfuscation | HoSNN |
|---|---|
| (1) Single-step attack performs better compared to iterative attacks | ✓ |
| (2) Black-box attacks perform better compared to white-box attacks | ✓ |
| (3) Increasing perturbation bound can't increase attack strength | ✓ |
| (4) Unbounded attacks can't reach 100% success | ✓ |
| (5) Adversarial example can be found through random sampling | ✓ |

Table 6: Checklist for gradient obfuscation

# E   More Experiment Data

Obfuscated gradients are a type of gradient masking that leads to a false sense of security when defending against adversarial examples Athalye et al. (2018); Carlini et al. (2019). Here we perform sanity checks including three obfuscated types and a checklist as per Athalye et al. (2018). First, we examine three types of obfuscated gradients. Specifically, we use the same surrogate to train HoSNN from scratch and deliver attacks; decent clean accuracy and a smooth training process indicate that it's not **Shattered Gradient** with nonexistent or incorrect value. As our defense doesn't introduce any random factors, **Stochastic Gradient** is not applicable. Our method also doesn't include any multiple iterations of neural network evaluation, so **Vanishing/Exploding Gradient** are also not applicable.

| Items to identify gradient obfuscation | HoSNN | Experiment |
|---|---|---|
| (1) Single-step attack performs better compared to iterative attacks | ✓ | Fig 5 and Table 8 |
| (2) Black-box attacks perform better compared to white-box attacks | ✓ | Fig 6 and Table 9 |
| (3) Increasing perturbation bound can't increase attack strength | ✓ | Fig 7 and Table 10 |
| (4) Unbounded attacks can't reach 100% success | ✓ | Fig 7 and Table 10 |
| (5) Adversarial example can be found through random sampling | ✓ | Fig 8 |

Table 7: Checklist for gradient obfuscation

| Dataset | Method | $\epsilon = 0$ | 2 | 4 | 6 | 8 | 16 | 32 | 64 |
|---|---|---|---|---|---|---|---|---|---|
| FMNIST | WB-FGSM | 92.31 | 90.26 | 88.16 | 86.25 | 84.70 | 77.70 | 60.70 | 24.27 |
| | WB-PGD | 92.31 | 89.34 | 85.44 | 80.61 | 74.91 | 47.93 | 5.87 | 0.00 |
| SVHN | WB-FGSM | 92.84 | 83.96 | 75.85 | 68.45 | 61.78 | 43.45 | 28.64 | 20.79 |
| | WB-PGD | 92.84 | 80.53 | 63.98 | 47.56 | 35.06 | 10.40 | 2.00 | 0.21 |
| CIFAR-10 | WB-FGSM | 90.00 | 77.84 | 71.14 | 67.05 | 63.98 | 56.66 | 48.92 | 26.93 |
| | WB-PGD | 90.00 | 72.24 | 58.66 | 49.65 | 42.63 | 24.38 | 6.82 | 0.65 |
| CIFAR-100 | WB-FGSM | 64.63 | 52.78 | 42.12 | 33.79 | 26.97 | 12.47 | 4.01 | 2.68 |
| | WB-PGD | 64.64 | 51.20 | 36.82 | 25.23 | 16.65 | 1.99 | 0.00 | 0.00 |

Table 8: HoSNN accuracy data for Test1. We compared the performance under white-box FGSM and PGD attack. Our data shows that single-step attacks are strictly weaker than multi-step attacks.

| Dataset | Method | $\epsilon = 0$ | 2 | 4 | 6 | 8 | 16 | 32 | 64 |
|---|---|---|---|---|---|---|---|---|---|
| FMNIST | BB-PGD | 92.31 | 91.05 | 89.81 | 88.85 | 87.67 | 84.61 | 75.12 | 39.93 |
| | WB-PGD | 92.31 | 89.34 | 85.44 | 80.61 | 74.91 | 47.93 | 5.87 | 0.00 |
| SVHN | BB-PGD | 92.84 | 89.31 | 86.10 | 82.68 | 78.49 | 57.02 | 25.96 | 8.02 |
| | WB-PGD | 92.84 | 80.53 | 63.98 | 47.56 | 35.06 | 10.40 | 2.00 | 0.21 |
| CIFAR-10 | BB-PGD | 90.00 | 86.55 | 83.46 | 80.11 | 76.61 | 56.77 | 13.43 | 0.92 |
| | WB-PGD | 90.00 | 72.24 | 58.66 | 49.65 | 42.63 | 24.38 | 6.82 | 0.65 |
| CIFAR-100 | BB-PGD | 64.64 | 61.68 | 59.22 | 56.67 | 54.06 | 42.08 | 16.90 | 1.82 |
| | WB-PGD | 64.64 | 51.20 | 36.82 | 25.23 | 16.65 | 1.99 | 0.00 | 0.00 |

Table 9: HoSNN accuracy data for Test2. We compared the performance under white-box PGD and black-box PGD attack. Our data shows that black-box attacks are strictly weaker than white-box attacks.

| Dataset | Method | $\epsilon = 0$ | 2 | 4 | 6 | 8 | 16 | 32 | 64 |
|---|---|---|---|---|---|---|---|---|---|
| FMNIST | SNN-WB-PGD | 92.92 | 78.87 | 58.03 | 42.27 | 30.54 | 2.71 | 0.00 | 0.00 |
| | HoSNN-WB-PGD | 92.31 | 89.34 | 85.44 | 80.61 | 74.91 | 47.93 | 5.87 | 0.00 |
| SVHN | SNN-WB-PGD | 95.51 | 44.78 | 9.66 | 1.83 | 0.44 | 0.10 | 0.02 | 0.01 |
| | HoSNN-WB-PGD | 92.84 | 80.53 | 63.98 | 47.56 | 35.06 | 10.40 | 2.00 | 0.21 |
| CIFAR-10 | SNN-WB-PGD | 92.47 | 44.00 | 11.73 | 2.46 | 0.56 | 0.01 | 0.00 | 0.00 |
| | HoSNN-WB-PGD | 90.00 | 72.24 | 58.66 | 49.65 | 42.63 | 24.38 | 6.82 | 0.65 |
| CIFAR-100 | SNN-WB-PGD | 74.00 | 11.62 | 1.29 | 0.13 | 0.04 | 0.00 | 0.00 | 0.00 |
| | HoSNN-WB-PGD | 64.64 | 51.20 | 36.82 | 25.23 | 16.65 | 1.99 | 0.00 | 0.00 |

Table 10: HoSNN accuracy data for Test3&4. We showed the performance of SNN and HoSNN under white-box PGD attack. Our data shows that increasing perturbation bound can increase attack strength and the accuracy can drop to 0 as the attack becomes stronger.

