# OpenReview forum: "HoSNNs: Adversarially-Robust Homeostatic Spiking Neural Networks with Adaptive Firing Thresholds"
_TMLR — Accepted by TMLR_

### Review · Reviewer_R1de · 2024-12-24

**Summary Of Contributions:**

This paper presents an advancement in improving the adversarial robustness of SNNs through a novel bio-inspired approach. The primary contribution is the introduction of the TA-LIF neuron model, which draws inspiration from biological homeostasis mechanisms. This model represents a fundamental departure from traditional defense methods, offering a biologically plausible solution to the challenge of adversarial attacks.

The authors develop HoSNNs that leverage these TA-LIF neurons, introducing an innovative concept called Neural Dynamic Signatures that serve as reference points for stabilizing neural activity. The theoretical foundation of the work is particularly strong, providing rigorous mathematical analysis of the TA-LIF neurons' dynamic properties, including bounded-input bounded-output stability and suppressed time growth of membrane potential error.

**Audience:**

Yes

**Broader Impact Concerns:**

N/A.

**Claims And Evidence:**

Yes

**Requested Changes:**

- Include a more intuitive explanation of the second-order homeostatic control system in Section 3.3.2, possibly with illustrative examples
- Expand the comparison in Section 4.5 to include more attack types
- Add concrete analysis of NDS storage requirements in Section 3.2

**Strengths And Weaknesses:**

**Strengths**:

The theoretical foundation of the work is exceptionally strong, presenting a comprehensive mathematical analysis that rigorously proves the stability and robustness properties of the proposed TA-LIF neurons. This analysis not only validates the approach but also provides valuable insights into why the method works.

The biological inspiration behind the work represents a novel perspective in adversarial robustness research. By drawing parallels between neural homeostasis and adversarial defense, the authors open up a new direction for developing robust neural networks that align with biological principles.

The empirical evaluation is particularly thorough, covering multiple datasets, architectures, and attack types. The authors' careful consideration of gradient obfuscation issues and comprehensive ablation studies demonstrate the reliability of their results. The significant performance improvements achieved across all test cases provide strong evidence for the method's effectiveness.

**Weaknesses**:
- In Section 3.1, the transition from equation (4) to the practical implementation using NDS could be better explained. The paper jumps from the theoretical formulation to the NDS solution without fully justifying why this approximation is valid or discussing potential limitations.

- In Section 3.3.2, the description of error minimization as a second-order homeostatic control system could benefit from more intuitive explanations. The connection between biological homeostasis and the mathematical formulation isn't immediately clear for readers without strong control theory background.

- The performance comparison in Section 4.5 only considers white-box FGSM and PGD7 attacks. A more comprehensive comparison including other attack types (like black-box attacks) would strengthen the evaluation, especially given the paper's claims about general robustness. The authors can just briefly describe other attack types.

- In Section 3.2, the storage overhead of maintaining Neural Dynamic Signatures isn't thoroughly analyzed. While the paper mentions it "remains manageable", concrete numbers or theoretical bounds would be more convincing.

---

> ### Author Response · Authors · 2025-01-05
> **Thank you for your helpful review!**
>
> Dear Reviewer, We sincerely appreciate your insightful and helpful comments. For your suggestions, we provide the following responses and clarifications:
>
> ## 1. Add concrete analysis of NDS storage requirements in Section 3.2
>
> Thanks to the author for the feedback. Experimentally, we recorded the equipment and time consumption in the appendix chapter D. We will add the following analysis of the time and space complexity of HoSNN in the main text:
>
> Assume that the number of neurons in the fully connected layer of the standard SNN is n, the stored weights are $w=O(n^2)$, and the training time is T.
>
> Additional space complexity: HoSNN needs to store the learnable parameters of each neuron $\theta_i= O(n)$. Compared with the network weights w~O(n^2), the storage cost is negligible.
>
> Additional time complexity:
> -Training stage: To train a HoSNN, we first need to train an SNN with the same architecture, which is the main source of extra time. According to our experiments, the extra computation time added by TA-LIF (learning of $\theta_i$ and dynamic calculation of $V_{th}$) is small, <10%. Therefore, the overall training time of HoSNN is ~2T.
>
> -Inference phase, the additional time brought by the dynamic threshold calculation observed in the experiment is negligible. The inference cost of a well-trained HoSNN is almost the same as that of an SNN.
>
> Potential Challenges:
>
> -$\theta_i$ initialization: In experiments, we observed that initializing $\theta_i$ too large will affect the stability of network training. When the initial value of theta_i is set to a large value (~1), the threshold of each neuron will change dramatically at the beginning of training, causing HoSNN to fail to converge. In experiments, we usually set $\theta_i$ between 0 and 0.5. More complex data sets usually require a smaller initial $\theta_i$.
>
> -Optimizer choice: HoSNNs usually have slower convergence in the later stages of training, especially in adversarial training. Our best practice is to use the Adam optimizer with a cosine decay learning rate. This can achieve similar convergence epochs as SNNs.

---

> ### Author Response · Authors · 2025-01-05
> **continued**
>
> ## 2.Expand the comparison in Section 4.5 to include more attack types
>
> Thank you for your reminder. We have added more experimental results in the table below and will revise it in the main text.
>
> | Dataset   | Method                                    | Clean   | FGSM    | RFGSM      | PGD(7)   | PGD(20)   | PGD(40)   |APGD(CE)  | APGD(DLR) | T-APGD  |
> |-----------|------------------------------------------|---------|---------|---------|---------|---------|---------|---------|---------|---------|
> | CIFAR-10          | Ding et al. (2022) [3]                 | 90.74   | 32.80   | 55.29       | 13.47   | 10.06      | 9.45      | 9.07      | 11.71       | 7.69      |
> |           | Özdenizci et al. (2023) [4]            | **92.14** | 33.94   | 57.52           | 15.98   | 13.18       | 12.67       | 10.05       | 14.07       | 9.19       |
> |           | Our work                                | 90.00   | **63.98** | **71.07**          | **43.33** | **40.97**       | **40.02**       | **38.89**       | **37.94**       | **41.69**       |
> | CIFAR-100            | Ding et al. (2022) [3]                 | 69.32   | 19.79   | 32.39          | 7.76    | 6.04       | 5.84       | 5.22      | 8.81       | 4.72      |
> |           | Özdenizci et al. (2023) [4]            | **70.02** | 18.60   | 32.55       | 9.53    | 7.64      | 7.43       | 6.62       | 8.88      | 5.41      |
> |           | Our work                                | 64.64   | **26.97** | **41.45**          | **16.66** | **15.83**      | **15.37**       | **12.55**       | **13.66**       | **10.2**      |

---

> ### Author Response · Authors · 2025-01-05
> **continued2**
>
> ## 3.NDS's approximation explaination and potential limitations.
>
> Thank the reviewer for pointing this out. We acknowledge that there is a logical gap between Eq(4) and the proposal of NDS. To alleviate this, we explain and will add the following paragraph to the text:
>
> $$
> \min E_i\left(x, x^{\prime}\right)=\sum_{i=0}^N \sum_{t=0}^T\left\|u_i\left(t | x^{\prime}\right)-u_i(t | x)\right\|^2
> $$
>
> Eq(4) describes an "ideal" optimization problem. Obviously, when $E_i(x, x')=0$, all neuron activities under adversarial sample input are the same as normal sample input. This can certainly achieve "adversarial robustness" in theory, but it is impossible to achieve in practice. There are two main limitations:
>
> 1. For a trained network, the intensity and type of attack are determined by external attackers, which means that x' has a huge range of variation.
> 2. For a trained network, the attack sample x cannot be determined in advance, which means that it is impossible to obtain an accurate $u_i(t|x) $.
>
> Under the constraints of these two problems, we still hope to adopt the idea of ​​Eq (4). A feasible approximation is that we use the available average value of the training data set $\mathbb{E}_{x \sim \mathcal{D}}\left[u_i(t|x)\right]$ as an approximation of the unavailable $u_i(t|x) $.  Such a method, as a 0th-order approximation, will bring about inaccurate limitations, similar to using the average feature of the "cluster center" to represent the "feature of each sample". We look forward to improving this in subsequent work, possibly by introducing an average NDS for each "category" or selecting a "reference sample" for each class of samples to calculate the NDS.

---

> ### Author Response · Authors · 2025-01-05
> **continued3**
>
> ## 4. A more intuitive explanation of the second-order homeostatic control system
>
> Thanks to the author for pointing this out. We would like to provide a more intuitive explanation of Eq(9) and Eq(10). Let's first review the equations. We have TA-LIF's second order dynamics under attack:
>
> $$
> \tau_m \frac{d^2 e_i(t)}{d t^2}+\frac{d e_i(t)}{d t}+r_i \tau_m \theta_i e_i(t) = \varepsilon(t)
> $$
>
> Noticed that when $ \theta_i = 0$,  according to  Eq(10), TA-LIF no longer has a dynamically changing threshold and thus degenerates into standard LIF. Its second-order dynamics degenerates into:
>
> $$
> \tau_m \frac{d^2 e_i(t)}{d t^2}+\frac{d e_i(t)}{d t} = \varepsilon(t)
> $$
>
> An example to intuitively understand these two formulas is that they correspond to a driven damped oscillator in the physical world, that is, the movement of a ball connected to a spring under a driving force and damping environment. Considering $e_i(t)$ as the distance the ball deviates from the equilibrium position, we can understand the physical meaning of each term in the formula:
>
> $\tau_m \frac{d^2 e_i(t)}{d t^2}$ describes the acceleration of the ball; $\frac{d e_i(t)}{d t}$ describes the frictional resistance of the ball;  $r_i \tau_m \theta_i e_i(t)$ describes the restoring force of the spring, which is the tendency of the spring to pull the ball toward the equilibrium position ($e_i(t) = 0$); $\varepsilon(t)$ describes the driving force exerted by the environment on the ball.
>
> With this explanation, we can easily understand the difference between the second-order dynamics of TA-LIF and LIF. The key term is $r_i \tau_m \theta_i e_i(t)$. For TA-LIF, it is like a small ball that is constantly pulled toward its initial equilibrium position by a spring when it is subjected to external disturbances: the result is that the ball will not deviate too far ($e_i(t)$ remains small). However, there is no such term in the dynamics of LIF, which causes $e_i(t)$ to become larger under external disturbances.

---

> ### Comment · Reviewer_9yb8 · 2025-02-13
> **An additional comment on another response**
>
> A comment on the response **2.Expand the comparison in Section 4.5 to include more attack types**:
>
> I believe the authors should have also re-visited this table for Reviewer R1de as well, given that they acknowledged the inappropriate experimental comparisons presented in Section 4.5. Different architectures and models are being compared across different attack settings, which is also the case here.

---

### Review · Reviewer_prg6 · 2024-12-24

**Summary Of Contributions:**

The paper a bio-inspired approach for SNNs’ adversarial robustness. The proposed approach is essentially based on an adaptive thresholding mechanism for the neurons. Specifically, the authors propose a new adaptive neuron, i.e., TA-LIF, which incorporates an online feedback control at inference time to adjust the firing threshold. The feedback is defined based on a hypothesis that the distribution of membrane potential for clean input is a reliable reference for benign behavior, and therefore, the activation is adjusted according to the difference between the actual membrane potential and the reference expectation of the potential on supposedly clean data.
The authors run adversarial attacks experiments, provide a bounding analysis and run experiments to show that the approach is not a gradient masking mechanism.

**Audience:**

Yes

**Broader Impact Concerns:**

No ethical concerns

**Claims And Evidence:**

Yes

**Requested Changes:**

- Clarify Feedback Threshold Adaptation
- Support or reframe hypothesis about $u^*$
- Discuss the computational overhead and potential challenges of the two-stage training process (conventional SNN training followed by training the modified architecture).
- Elaborate on why adaptive thresholding at the node level inherently leads to robustness. Compare and contrast this with other bounded activation functions, such as sigmoid, which do not inherently offer robustness.

**Strengths And Weaknesses:**

Strengths:

+ The paper overall is well written.
+ The idea of adaptive thresholding for adversarial robustness is novel –
+ The adversarial attack experiments support the authors claim, and their methodology is sound.

Weaknesses:

-	It is unclear to me how the feedback threshold adaptation is performed
-	I think the authors lightly affirm that the u* is a reliable reference for benign (as opposed to adversarial) behavior. This is not an obvious fact, quite the opposite. In fact, if that’s the case, the adversarial robustness problem would have been fundamentally solved by detecting OOD in the features space. I encourage the authors to either: \textbf{(i)} reword their claims to frame this as a hypothesis not “expected” as in Section 3.2. \textbf{(ii)} Or, run experiments to support this claim, like for example test if this OOD w.r.t. a reference NDS explicitly enables detecting adversarial examples. This should be the case if their assumption/hypothesis is correct.
-	The approach technically involves 2 stage training: first, a training of a conventional SNN with a fixed threshold, and then, a new training for their modified architectures to re-adapt the parameters, and learn their new parameter theta. The authors should run experiments or at least discuss the computational overhead, and  training difficulty of their approach.
-	The authors propose an analysis of the bounds forced by their adaptive firing at node level, which makes sense. I am unsure why would adaptive thresholding behavior at node level lead to any inherent robustness. Sigmoid activation function offers bounds at neuron level, yet, it does not offer any inherent robustness. Could the authors discuss this point?
Minor :
-	Figure 3 – the subfigures are not properly labeled (a) and (b)
-	In Section 3.1., the expression of the adversarial noise is not accurate. The adversarial noise is generally an additive noise not multiplicative: $\delta.x $ -- and multiplying this by $\epsilon$ does not make it a budget. Instead: $x’=x+ \delta$, s.t. $\delta < \epsilon$

---

> ### Author Response · Authors · 2025-01-05
> **Thank you for your helpful review!**
>
> Dear Reviewer,
> We sincerely appreciate your insightful and helpful comments. For your suggestions, we provide the following responses and clarifications:
>
> ## 1. Clarify Feedback Threshold Adaptation
>
> Thank you for pointing out our oversight. We will add the following paragraph to the main text to better explain and clarify the implementation of threshold adaptation:
>
> Specifically, to train a HoSNN and implement the threshold adaptation mechanism, there are three stages:
>
> ### Step1. Obtaining NDS
>
> Train a standard LIF-based SNN on a clean training set and obtain the average neural activity (NDS). The NDS we obtain is the set of membrane potential time series of each neuron under the average of the data set. There is no threshold adaptation mechanism at this stage, and any well-trained LIF-SNN can be used to obtain NDS.
>
> ### Step2. Training of HoSNN
>
> Use NDS to train a HoSNN with TA-LIF. The dynamic threshold calculation mechanism occurs at **each sample $x$, each neuron $i$, and each time step $t$.** Assume that the input sample $x$ is given and the behavior of neuron $i$ is observed:
>
> - **Forward propagation stage**:
>   Calculate the neuron membrane potential $u_i(x, t)$. According to NDS $u^*_i(t)$, the current membrane potential $u_i(x, t)$, the sensitivity parameter $\theta_i$ of the neuron, and the following equation:
>
> $$
> V^i_{th}(x, t+1) =  V^i_{th}(x, t+1)  + \Delta t*\theta_i * [u_i(x, t) - u^*_i(t)]
> $$
>
> The dynamic threshold $V^i_{th}(x, t)$ is calculated. At this stage, the threshold adaptation mechanism is turned on by default.
>
> - **Back propagation stage**:
>   According to the gradient information, update the network connection weights $w$ and the neuron sensitivity parameters set $\theta_i$ (different for each neuron). The learning of $\theta_i$ is designed to fully consider the different degrees of adaptation of each neuron to changes in external input.
>
> ### Step3. HoSNN Inference:
>
> When a well-trained HoSNN has been obtained, including $w$ and $\{\theta_i\}$, the threshold adaptation mechanism is turned on by default for any sample reasoning (adversarial or non-adversarial). As in step 2, the calculation of the dynamic threshold $V^i_{th}(x, t)$ occurs at each sample $x$, each neuron $i$, and each time step $t$ according to the same equation. When there is an adversarial sample input, the dynamic threshold can filter out some "abnormal neural activity"—that is, abnormal values of the membrane potential—in an adaptive way, thereby reducing the output spikie deviation of the neuron.

---

> ### Author Response · Authors · 2025-01-05
> **Continued**
>
> ## 2.Support or reframe hypothesis about $u^*$
>
> Thank you for pointing out our oversight. We agree with your point of view: In SNN, whether the membrane potential sequence can be used as an effective solution to the adversarial attack problem is still an unclear issue. As far as we know, there is no relevant research. We will rephase the main text and point it out as a hypothesis: We assume that in SNN $u^*$ (i.e., the original neural activity in the training set), similar to the neuron activation value in ANN, is of reference significance for adversarial perturbations.
>
> Many existing studies in ANN support this hypothesis: detection in feature space can alleviate adversarial problems. We will add the following papers to the main text citations:
>
> In (1), the author proposed a loss, which is the difference of topk neural activities under adversarial sample and clean sample input. Experiments show that optimizing this additional loss effectively increases the robustness of ANN. Similar to the core idea of ​​this paper, we minimize the difference of neural activities under attack by threshold adaptation.
>
> In (2) The authors show that in neural networks, adversarial attacks mainly propagate through a few critical paths. The attacked neural activity along these pathways deviates significantly from the original neural activity. Similar to the above, the author found that by reducing the sensitivity of loss to the output of these neurons, the overall robustness of the network can be effectively improved.
>
> In (3), The author proposes that the adversarial vulnerability of neural networks comes from the inherent "non-robust" features in the dataset, and adversarial attacks fully exploit and amplify these features. The model's reliance on these non-robust features creates adversarial attack vulnerability. In this sense, our method can be seen as a passive reduction of the network's sensitivity to learned features, where the learned $\theta_i$ represents the sensitivity of a single neuron. (In a sense, as the reviewer worried, NDS does contain certain non-robust features. But in terms of attacks, the features of adversarial attacks can be regarded as a signal of OOD compared with the features of standard samples.)
>
> In(4), the author propose to augment deep neural networks with a small "detector" subnetwork which is trained on the binary classification task of distinguishing genuine data from data containing adversarial perturbations. The detector take the main network's    intermediate features to distinguish whether the input is an adversarial sample. Similar to what the reviewers expected, we can indeed use the intermediate features of the network to detect adversarial examples in an OOD manner. In addition, there is a line of work in the field of adversarial defense to identify adversarial samples by detecting intermediate features. See (5).
>
> (1) C. Zhang et al., "Interpreting and Improving Adversarial Robustness of Deep Neural Networks With Neuron Sensitivity," in IEEE Transactions on Image Processing, vol. 30, pp. 1291-1304, 2021, doi: 10.1109/TIP.2020.3042083.
>
> (2)Tianlin Li, Aishan Liu, Xianglong Liu, Yitao Xu, Chongzhi Zhang, Xiaofei Xie,
> Understanding adversarial robustness via critical attacking route,
> Information Sciences, Volume 547, 2021.
>
> (3)A. Ilyas et al. Adversarial Examples Are Not Bugs, They Are Featuress. 2019.
>
> (4)On Detecting Adversarial Perturbations. ICLR 2017.
>
> (5)Opportunities and Challenges in Deep Learning Adversarial Robustness: A Survey.

---

> > ### Comment · Reviewer_prg6 · 2025-01-17
> > **Even in ANN, this hypothesis is not obvious**
> >
> > The question of potentially observing adversarial noise in ANNs and detecting attacks has been investigated under the hypothesis the authors are mentioning, I agree.
> > It has been also strongly refuted. See the 2 papers below:
> > [1] Nicholas Carlini and David Wagner. 2017. Adversarial Examples Are Not Easily Detected: Bypassing Ten Detection Methods. In Proceedings of the 10th ACM Workshop on Artificial Intelligence and Security (AISec '17). Association for Computing Machinery, New York, NY, USA, 3–14. https://doi.org/10.1145/3128572.3140444
> > [2] Tramer, F.. (2022). Detecting Adversarial Examples Is (Nearly) As Hard As Classifying Them. Proceedings of the 39th International Conference on Machine Learning, 162:21692-21702 Available from https://proceedings.mlr.press/v162/tramer22a.html.

---

> ### Author Response · Authors · 2025-01-05
> **Continued2**
>
> ## 3.Discuss the computational overhead and potential challenges of the two-stage training process
>
> Thanks to the author for the feedback. Experimentally, we recorded the equipment and time consumption in the appendix chapter D. We will add the following analysis of the time and space complexity of HoSNN in the main text:
>
> Assume that the number of neurons in the fully connected layer of the standard SNN is n, the stored weights are $w=O(n^2)$, and the training time is T.
>
> Additional space complexity: HoSNN needs to store the learnable parameters of each neuron $\theta_i= O(n)$. Compared with the network weights w~O(n^2), the storage cost is negligible.
>
> Additional time complexity:
> -Training stage: To train a HoSNN, we first need to train an SNN with the same architecture, which is the main source of extra time. According to our experiments, the extra computation time added by TA-LIF (learning of $\theta_i$ and dynamic calculation of $V_{th}$) is small, <10%. Therefore, the overall training time of HoSNN is ~2T.
>
> -Inference phase, the additional time brought by the dynamic threshold calculation observed in the experiment is negligible. The inference cost of a well-trained HoSNN is almost the same as that of an SNN.
>
> Potential Challenges:
>
> -$\theta_i$ initialization: In experiments, we observed that initializing $\theta_i$ too large will affect the stability of network training. When the initial value of theta_i is set to a large value (~1), the threshold of each neuron will change dramatically at the beginning of training, causing HoSNN to fail to converge. In experiments, we usually set $\theta_i$ between 0 and 0.5. More complex data sets usually require a smaller initial $\theta_i$.
>
> -Optimizer choice: HoSNNs usually have slower convergence in the later stages of training, especially in adversarial training. Our best practice is to use the Adam optimizer with a cosine decay learning rate. This can achieve similar convergence epochs as SNNs.

---

> ### Author Response · Authors · 2025-01-05
> **Continue3**
>
> ## 4.Corrected minor errors in figure 3 and section 3.1
>
> We thank the reviewer for pointing out our typo. We will fix the typo of Figure 3 and Section 3.1 in the main text.
>
> ## 5. Elaborate on adaptive thresholding's robustness and Compare with bounded activation functions
>
> We thank the reviewer for his insightful comments. We will make the following explanation and add it to the main text:
>
> The robustness of TA-LIF does not come from bounded activation functions, but from its dynamic stability. Let's start with the sigmoid function: for inputs x and x', after sigmoid activation, they become $\sigma(x)$ and $\sigma(x')$ in 0,1. Even though sigmoid is a bounded activation function, $\sigma(x)  != \sigma(x')$ .
>
> In standard SNNs, the LIF neuron is similar. If LIF, i.e., postsynaptic potential, is considered as an activation function, when input current $ x != x'$, usually LIF(x) != LIF(x'). Sensitivity to input changes is an important property, otherwise it will lead to zero gradient of the activation function.
>
> But in the context of adversarial attacks, we want to minimize the difference between TALIF(x) and TALIF(x'). Because at the neuron level, the closer each TALIF(x') is to TALIF(x), the closer the final network output classification result y(x') is to y(x), which implies adversarial robustness.
>
> Thanks to the dynamic characteristics of LIF neurons, we achieve this through carefully constructed dynamic thresholds, that is, although we cannot strictly achieve TALIF(x') = TALIF(x), we can use dynamic thresholds to make TALIF(x') $\rightarrow$ TALIF(x). Specifically, substituting Eq(10) into Eq(9), we can get the second-order differential equation for the error $e_i(t)$
>
> $$
> \tau_m \frac{d^2 e_i(t)}{d t^2}+\frac{d e_i(t)}{d t}+r_i \tau_m \theta_i e_i(t)=\varepsilon(t)
> $$
>
> This is a damped oscillation equation. When the driving term on the right side of the equation is small, $e_i(t)$ will decay from the initial value to a value close to 0 over time. The detailed theoretical analysis is provided by Eq(11, 12, 13) and Appendix B, C. The experimental visualization can be mainly referred to figure 3. By reducing $e_i(t)$, we make TALIF(x')  $\rightarrow$ TALIF(x), and thus provide a certain degree of network robustness.

---

> ### Author Response · Authors · 2025-01-29
> **Thanks**
>
> Dear reviewer:
> Thank you for providing these references. After reading, we understand and agree your concerns. At this stage, the point that "clean neural activity could contributes to the robustness of SNNs" should be proposed as an explicit hypothesis until it is widely tested by the community and a consensus is reached. We will revise and clarify this point in the main text.
>
> Here, we provide an intuitive understanding of NDS's effectiveness: for any attack sample $x' = x + \delta x$, as the strength of its adversarial attack decreases ($\delta x \rightarrow 0$), the sample gradually returns to the original clean sample ($x' \rightarrow x$). Hence the neural activity of the attacked network $f(x')$ will gradually return to the clean neural activity $f(x)$, i.e. $f(x') \rightarrow f(x)$. Then the classification result of the network will also return to the correct result ($y' \rightarrow y$). Therefore, if we can find a way to alleviate the difference between the neural activity of the attacked network $f(x')$ and the clean neural activity $f(x)$,  we can also mitigate the consequences of adversarial attacks.

---

### Review · Reviewer_9yb8 · 2025-01-17

**Summary Of Contributions:**

Authors propose homeostatic spiking neural networks (HoSNNs) as a novel category of bio-inspired SNNs that exploit threshold-adapting LIF neurons, to improve SNN robustness against adversarial attacks. Specifically, each neuron in the resulting adversarially trained HoSNN exploits dynamically adapting firing thresholds via a temporal neural dynamic signature anchor that is estimated using clean data on a shadow SNN with identical architecture. By adapting neuron-specific firing thresholds based on this anchor, the proposed model aims to minimize the membrane potential errors of all neurons, which are estimated by the difference in membrane potentials in response to benign versus adversarial examples. Empirical simulations on Fashion-MNIST and CIFAR-10/100 type of datasets show that the proposed method is effective when combined with adversarial training.

**Audience:**

Yes

**Claims And Evidence:**

No

**Requested Changes:**

- Current "Section 4.5 - Comparison with other works" should be revised thoroughly from scratch. Table 5 is currently filled out by retrieving exact evaluation accuracies from the compared papers [Ding et al. 2022; Ozdenizci & Legenstein 2023], without carefully considering the details of these evaluations (e.g., SNN architecture, attack configuration and strength, etc.). Given the training/evaluation settings described in Section 4.1 of this submission, authors currently demonstrate a misleading benchmarking comparison of different methods. To describe: this method uses a VGG-5 for CIFAR-10 with T=5. This is a different architecture than all other methods that are compared, which actually use VGG11 architectures with more parameters, and a longer T=8 simulation length. These differences in fact make the models more vulnerable to attacks already. Furthermore, the FGSM and PGD7 attack evaluations that are retrieved from those works, in fact correspond to significantly stronger _ensemble SNN attacks_, where the surrogate gradient used by the adversary can rigorously adapt for each attacked sample. Authors' method does not appear to be evaluated in this way. Therefore, the methods are not compared on the same grounds either, which is a critical benchmarking step in empirical adversarial robustness studies. Basically, to be fair and rigorous, empirical adversarial robustness comparisons with other methods should be always performed on the same grounds, using the same architectures, and exact same adversarial attack configurations.

- Could you discuss or empirically demonstrate how well the proposed approach generalizes to even slightly larger ML tasks, such as TinyImageNet classification, or residual CNN architectures?

- Can the authors discuss why item (1) of the checklist for gradient obfuscation does not actually hold in their black-box attack evaluations? As far as I can see, multi-step attacks somehow perform worse than single-step attacks in the results presented in Table 3. Can the authors re-run these experiments on different seeds to generate their black-box transfer attacks?

- In Section 2.1, authors also mention RGA [Bu et al. 2023], and HART [Hao et al. 2024] attacks specifically tailored for SNNs. However the paper does not evaluate the robustness of the proposed SNN against these at all, but with ANN-like attacks. Can the authors discuss why this is the case, and if it is a limitation?

**Strengths And Weaknesses:**

Strengths:
- Paper is written clearly and has a good motivating narrative that well illustrates the idea.
- Proposed idea of using bio-inspired homeostasis for adversarial robustness in SNNs is a novel approach.

Weaknesses:
- Empirical evaluations of the defense and benchmarking appears problematic in some parts, which is important from the perspective of securely/robustly solving machine learning tasks with the proposed SNN architecture.

---

> ### Author Response · Authors · 2025-01-30
> **Thank you for your helpful review!**
>
> Dear Reviewer, We sincerely appreciate your insightful and helpful comments. For your suggestions, we provide the following responses and clarifications:
>
> ## 1.Inappropriate experimental comparison.
>
> We appreciate the reviewer pointing out the unaligned comparison in our paper. We  re-run the experiments in section 4.5 to get the same experimental setup as the reference and make a fair comparison.
>
> The core reason why we use Vgg7 instead of a larger network Vgg11 is that we observed performance saturation: for example, on the cifar10 and cifar100 datasets, Vgg7 has achieved the same clean accuracy as the reference papers (~92.5% on cifar10; ~74% on cifar100), and it is not necessary to increase the network and increase the computation time. However, we will still add some experiments for fair comparison.
>
> On the other hand, we provide detailed ablation experiments in sections 4.2, 4.3, and 4.4. The results of these ablation experiments are the core evidence of the effectiveness of HoSNN. We believe that the different experimental settings does not become a core factor affecting the effectiveness of our claim method.
>
> Experimentally, we added more experimental data below:
>
> ### Table: Comparison with others work
>
> | Data       | Methods                                | Net  | Train          | Attack | Clean       | FGSM        | PGD7        |
> |------------|---------------------------------------|------|----------------|--------|-------------|-------------|-------------|
> | **CIFAR-10** | [Ding et al., 2022](#ref-ding2022snn)              | VGG7 | $\epsilon = 2$ | BPTT   | 83.45       | 39.69       | 20.14       |
> |            | [Ozdenizci et al., 2023](#ref-ozdenizci2023adversarially) | VGG7 | $\epsilon = 2$ | BPTT   | **91.86**   | 41.55       | 27.35       |
> |            | *Our work*                            | VGG7 | $\epsilon = 2$ | BPTT   | 90.00       | **63.98**   | **42.63**   |
> | **CIFAR-100** | [Ding et al., 2022](#ref-ding2022snn)              | VGG7 | $\epsilon = 4$ | BPTT   | **67.47**       | 25.38     | 15.66        |
> |            | [Ozdenizci et al., 2023](#ref-ozdenizci2023adversarially) | VGG7 | $\epsilon = 4$ | BPTT   | 67.26   | 21.35       | 13.45       |
> |            | *Our work*                            | VGG7 | $\epsilon = 4$ | BPTT   | 64.64       | **26.97**   | **16.66**   |
>
> **Note:** The bold numbers indicate the best performance in each category.
>
> We used the same network architecture and the same adversarial training intensity (FGSM, eps = 2 for cifar10, eps = 4 for cifar100). Experiments show that with the same network architecture, our results still outperform the latest references and are also the strongest baselines. We hope this result addresses your concerns.

---

> ### Author Response · Authors · 2025-01-30
> **Continue**
>
> ## 2.generalizes to even slightly larger ML tasks
>
> Thanks for your comments.
>
> **Theoretically**, the core of our approach is to build a neuron activation function with adaptive capabilities. There is no obvious reason to prevent the dynamic threshold method from being extended to larger networks. Following is the time and space complexity analysis of the proposed algorithm. Assume that the number of neurons in the fully connected layer of the standard SNN is n, the stored weights are $w=O(n^2)$, and the training time is T.
>
> **Additional space complexity**: HoSNN needs to store the learnable parameters of each neuron $\theta_i= O(n)$. Compared with the network weights w~O(n^2), the storage cost is negligible.
>
> **Additional time complexity**:
>
> -Training stage: To train a HoSNN, we first need to train an SNN with the same architecture, which is the main source of extra time. According to our experiments, the extra computation time added by TA-LIF (learning of $\theta_i$ and dynamic calculation of $V_{th}$) is small, <10%. Therefore, the overall training time of HoSNN is ~2T.
>
> -Inference phase, the additional time brought by the dynamic threshold calculation observed in the experiment is negligible. The inference cost of a well-trained HoSNN is almost the same as that of an SNN.
>
>
> **Experimentally**, we train a small residual CNN on TinyImageNet classification for SNN and HoSNN. Its architecture is conv64-conv128-conv256-resblock1-reblock2-resblock3-fc4096-fc4096-fc200. Each resblock contains 2 Conv512 layer. Each parameterized layer is followed by LIF/ALIF neurons as activation functions. Adversarial training is performed at 2/255 and testing is performed at 4/255.
>
> | Data       | Methods                                | Net  | Train          | Attack | Clean       | FGSM        | PGD7        |
> |------------|---------------------------------------|------|----------------|--------|-------------|-------------|-------------|
> | **TinyImageNet** | SNN             | ResCNN | x | BPTT   | 54.10 | 1.68 | 0.00     |
> |  | HoSNN| ResCNN | $\epsilon = 2$ | BPTT   | 50.14       |  20.18       | 15.72       |
>
> Experimental results show that our method is still effective on ResCNN+TinyImageNet.

---

> ### Author Response · Authors · 2025-01-30
> **Continue2**
>
> ## 3. Explain Table 4 item (1)  and Table 3's Black box result. ##
>
> We thank the reviewer for his insightful analysis. We also observed that in the experimental results of black-box attacks in Table 3, it is true that multi-step attacks (PGD/BIM) are slightly less powerful than single-step attacks (FGM). However, this does not conflict with the first item in Table 4, "Single-step attack performs better compared to iterative attacks". We provide the following explanation:
>
> The core reason is: we use a vanilla SNN with similar architecture and trained independently with different seeds to generate black-box adversarial examples. The effectiveness of single-step and multi-step adversarial examples can be demonstrated by the fact that on the original SNN, all generated black-box are stronger in multi-step than in single-step (Table 1 & 2). When using the same black-box dataset to attack HoSNN, SNN/HoSNN showed stronger resistance to some multi-step attacks, which may be due to its inherent robustness rather than invalid black-box attack samples. A possible explanation for FGSM to be more effective than PGD in HoSNN's black-box attacks: PGD could generate stronger out-of-distribution semantic information than FGSM, triggering more drastic threshold changes, which could in turn suppress the larger PGD's perturbations by a higher degree. Hence this mechanism does not necessarily provide black-box robustness in a strictly monotonic way.
>
> In the context of white-box attacks, we conduct a detailed inspection of Table 4 item (1). See Fig 5 and Table 8 in the original text. All experimental results show that multi-step attacks are stronger than single-step attacks. We also added more black-box attack data under different seeds.
>
> | Dataset        | Net         | Clean      | FGSM       | RFGSM               | PGD7               |
> |----------------|-------------|------------|------------|---------------------|--------------------|
> | **Fashion MNIST** | WBSNN       | **92.92**  | 17.78      | 31.50 ± 0.04        | 0.00 ± 0.00        |
> |                | ×           | 92.08      | 66.26      | 78.44 ± 1.66        | 77.17 ± 3.09       |
> |                | ✓           | 92.31      | **68.31**  | **80.75 ± 0.05**    | **79.07 ± 0.95**   |
> | **SVHN**       | WBSNN       | **95.51**  | 10.20      | 8.82 ± 0.08         | 0.12 ± 0.03        |
> |                | ×           | 93.85      | 17.37      | 42.66 ± 0.02        | 29.36 ± 8.21       |
> |                | ✓           | 92.84      | **19.08**  | **44.49 ± 0.08**    | **35.80 ± 9.83**   |
> | **CIFAR-10**   | WBSNN       | **92.46**  | 7.56       | 0.89 ± 0.11         | 0.00 ± 0.00        |
> |                | ×           | 91.87      | 13.48      | 9.02 ± 0.23         | 0.15 ± 0.04        |
> |                | ✓           | 90.00      | **25.18**  | **31.22 ± 0.11**    | **16.36 ± 2.94**   |
> | **CIFAR-100**  | WBSNN       | **74.00**  | 2.57       | 0.12 ± 0.01         | 0.00 ± 0.00        |
> |                | ×           | 68.72      | 12.18      | 17.88 ± 0.01        | 8.26 ± 1.47        |
> |                | ✓           | 64.64      | **14.54**  | **24.33 ± 0.01**    | **17.82 ± 0.92**   |
>
> **Table:** SNNs (×) and HoSNNs (✓) black-box attack accuracy, trained with FGSM adversarial training and tested by different black-box attack methods with ϵ = 32/255. The WBSNN row is the attack result of the black-box data used for testing on the SNN that originally generated it, that is, the white-box attack with ϵ = 32/255 on the original SNN. We used 5 different seeds to generate different black-box attack samples and show the results. Since there is no randomness in the FGSM attack, the results will not change.
>
> According to the results, all multi-step attacks are stronger than single-step attacks on the original network. Therefore, they are effective black-box attack samples. On the large-scale datasets CIFAR10 and CIFAR100, all black-box multi-step attacks are stronger than single-step attacks. Therefore, they follow item (1) of the checklist.
>
> On the small-scale datasets Fashion MNIST and SVHN, we speculate that a few multi-step attacks are not as good as single-step attacks due to the inherent robustness of SNN/HoSNN, especially the adaptive mechanism of HoSNN does not necessarily provide monotonic defense capabilities.

---

> ### Author Response · Authors · 2025-01-30
> **Continue 3**
>
> ## 4.Missing SNN-specific attacks. ##
>
> Thanks to the reviewer for the suggestion. We did not include them mainly because we wrote this paper from the perspective of the defender rather than the attacker. To date, attacks such as FGM, PGD, and APGD are still recognized as effective methods in the ML security community, both in SNN/ANN. In the SNN defense references we cited, almost all related works only consider the defense against the above "famous attacks". There are some reasons for this:
>
> On the one hand, the SNN field has not yet formed a recognized basic parameter setting similar to that of ANN, which makes it difficult for new attack methods to be quickly transferred to different SNN settings. There are a large number of neuron hyperparameters in SNN (such as membrane potential time constant, membrane voltage, synaptic time constant); neuron structure (such as IF/LIF, 0/1 order synapse, etc.); network training method (ANN conversion, BPTT, proxy gradient); information encoding method (rate, spike train, spike time). Usually, specific attacks on SNNs rely on a specific network setting, and additional efforts are needed to extend them to other SNNs. For example, RGA relies on the encoding method assumption of rate coding in SNN, but we use spike train coding. In HART, the authors used a 0th order postsynaptic structure, but we used a 1st order postsynaptic structure. These specific setting differences prevent us from using existing SNN attack methods.
>
> On the other hand, new attack methods emerge in an endless stream. Although important, it is difficult for defenders to cover all recently published attack methods. We acknowledge that this is an important practice, but many newly proposed methods still need to be tested by the SNN ML security community.

---

> ### Author Response · Authors · 2025-02-13
> **Continue**
>
> Dear reviewer,
> We have added additional experiments and discussions as you requested. Please let us know if you have further questions.
> Thanks,
> Authors

---

> > ### Comment · Reviewer_9yb8 · 2025-02-13
> > **Response to authors' rebuttal**
> >
> > Thanks to the authors for their responses and efforts.
> >
> > - Thanks for the corrections to the **1. Inappropriate experimental comparisons**. All these changes should be also reflected in the revised manuscript, with clear descriptions of the implementations. One question to clarify regarding the associated table above: Are these listed FGSM and PGD7 attacks performed with a surrogate gradient ensemble attack approach (which is the state-of-the-art SNN robustness evaluation method from [1])? Or is the attack evaluation approach different than before in this table? [1] “Adversarially robust spiking neural networks through conversion” TMLR 2024.
> >
> > - Regarding the experiments on **2. generalizing to larger ML tasks**: The posted TinyImageNet results table compares a vanilla trained SNN, with an adversarially trained HoSNN. The demonstrated benefit might as well be due to the adversarial training?
> > Based on this presentation, authors' claim that HoSNN is a scalable way to achieve a better robustness-accuracy tradeoff is not well justified. One should actually show HoSNN without adversarial training, or an SNN with e=2 adversarial training, in these comparisons.

---

> ### Author Response · Authors · 2025-02-14
> **Response to Reviewer**
>
> Dear reviewer,
>
> We've updated the revised pdf, with the modified parts shown in blue.
>
> For Q1, for fair comparison, we use standard fgsm and pdf for all three papers. We do not use any of the stronger ensemble attacks in [1]. It's mainly because our method does not involve the conversion from ANN to SNN, so the gradient when directly training SNN is considered as the preferred attack scheme.  It demonstrates the additional gain of HoSNN for adversarial training.
>
> For Q2, we supplement the latest robustness results of SNN and HoSNN under same FGSM $\epsilon =2$ adversarial training.
>  | Data       | Methods                                | Net  | Train          | Attack | Clean       | FGSM        | PGD7        |
> |------------|---------------------------------------|------|----------------|--------|-------------|-------------|-------------|
> | **TinyImageNet** | SNN             | ResCNN | $\epsilon = 2$ | BPTT   | 50.32       | 10.29       | 4.95       |
> |  | HoSNN| ResCNN | $\epsilon = 2$ | BPTT   | 45.68       |  **19.54**       | **15.58**       |

---

> > ### Comment · Reviewer_9yb8 · 2025-02-19
> > **Thanks for the responses**
> >
> > Thanks to the authors for their efforts and further clarifications. I now see the revisions to the manuscript, and it has been greatly improved.

---

### Author Response · Authors · 2025-02-10
**Thank you for your helpful review!**

Dear Reviewers and Action Editors,
We have provided responses to all reviewers’ questions and resubmitted the revised pdf. If you have more questions, please let us know.

Thanks,
Authors

---

> ### Author Response · Authors · 2025-02-14
> **Updated revised version**
>
> Dear Reviewers and Action Editors,
>
> We have updated the revised pdf as requested by all reviewers, with the modified parts shown in blue.
> Thanks,
> Authors

---

### Decision · Action_Editor_8YNF · 2025-02-14

**Recommendation:** Accept as is

**Comment:**

The paper introduces a novel mechanism to improve adversarial robustness of SNNs. They provide a rigorous mathematical foundation and solid empirical evidence. Reviewer comments were adequately addressed. The work is of somewhat limited scope and does not present a significant improvement over the state of the art for machine learning tasks.

**Audience:**

Yes, the manuscript is interesting for researchers working in the field of spiking neural networks and for an audience interested adversarial robustness.

**Claims And Evidence:**

Yes.